# LONG-TAILED LEARNING WITH MUON OPTIMIZER

## ABSTRACT

Long-tailed recognition poses a significant challenge in deep learning, as models tend to be biased towards head classes, leading to poor generalization on underrepresented tail classes. A key factor contributing to this issue is that the optimization process for tail classes often stalls in sharp regions of the loss landscape. In this work, we investigate this problem from an optimization perspective and leverage the recently proposed Muon optimizer. We provide new theoretical insights, demonstrating that Muon's gradient orthogonalization enhances the update's projection along directions of negative curvature, thereby facilitating a more effective escape from sharp minima. To further mitigate the additional computational overhead of Muon, we propose Progressive Muon Optimizer (ProMO), a novel hybrid optimization approach that balances performance with efficiency. Specifically, ProMO employs a sinusoidal probability schedule to dynamically alternate between SGD and Muon. This method predominantly uses computationally efficient SGD in the early stages of training and gradually increases the use of Muon as the model approaches convergence when escaping sharp minima becomes critical for tail-class generalization. Extensive experiments on large-scale long-tailed benchmarks demonstrate that ProMO consistently outperforms existing long-tailed recognition methods. These results validate that ProMO effectively improves generalization on tail classes without incurring significant computational costs, highlighting its potential as a practical and effective solution for long-tailed learning.

## 1 INTRODUCTION

Deep learning has significantly advanced a wide range of domains, from computer vision to large language models, achieving unprecedented performance largely driven by large-scale, high-quality datasets (Russakovsky et al., 2015). However, modern real-world datasets are often imbalanced, especially in domains such as medical diagnosis, where data collection is costly and time-consuming (Buda et al., 2018). In these fields, datasets typically exhibit long-tailed distributions, with a small number of dominant classes (head classes) being overrepresented, while others (tail classes) are significantly underrepresented. This class imbalance presents significant challenges during model training, as traditional learning algorithms tend to bias towards the head classes, leading to poor generalization for the tail classes (Wang et al., 2023). As a result, it has become crucial to explore robust training methods that can effectively handle long-tailed class distributions.

Many excellent methods have been proposed to address class imbalance, including re-sampling (Chawla et al., 2002), decoupling (Kang et al., 2020), loss rebalancing (Ma et al., 2023), and contrastive learning techniques (Zhu et al., 2022; Du et al., 2024). While these methods aim to alleviate the dominance of head classes, they often overexpose the limited tail class samples, thereby increasing the risk of overfitting. Recent studies (Rangwani et al., 2022; Li et al., 2025) have also shown that for minority classes in imbalanced datasets, the optimization process often converges to sharp regions in the loss landscape, characterized by large eigenvalues in the Hessian matrix, resulting in poor generalization performance for these underrepresented classes. One promising direction to address this issue is Sharpness-Aware Minimization (SAM) (Foret et al., 2021), a technique that focuses on escaping sharp minima by finding sharp maximal points in the neighborhood of the current weight and then minimizing the loss at these points. While SAM has been shown to improve generalization by helping the model escape saddle points, it comes at the cost of significantly increased training time, as it requires twice the number of backpropagation steps (Luo et al., 2024). This scaling issue poses significant challenges for applying SAM to large-scale datasets and models, where training time and computational efficiency are crucial considerations.

To overcome these challenges, we turn to the recently proposed Muon optimizer (Jordan et al., 2024; Shen et al., 2025), which modifies the SGD optimizer by orthogonalizing the gradient updates through Newton-Schulz iteration. We demonstrate that Muon effectively enhances the gradient component along the negative curvature, allowing the optimizer to converge to flatter regions of the loss landscape more efficiently, leading to improved generalization performance. This is especially crucial in the context of imbalanced datasets, where Muon helps to boost the performance of tail classes.

To further balance training cost with the benefits of Muon, we introduce a novel dynamic optimizer selection method, termed as ProMO. This approach uses a sinusoidal function to dynamically control the probability of selecting the Muon optimizer during the training process. As the model nears convergence, the probability of selecting Muon increases, providing continued support for escaping saddle points without excessively increasing the computational burden. Our contributions are summarized as follows:

1. We provide new theoretical insights into Muon from the perspective of loss landscape. Specifically, we show that Muon enhances the gradient component along the negative curvature, facilitating effective escape from sharp regions toward flatter minima. This is particularly crucial in long-tailed learning, where tail classes often converge to sharp minima, resulting in reduced generalization.

2. We propose a novel method, ProMO, to dynamically balance training cost and performance. By controlling the probability of using Muon through a sinusoidal schedule, ProMO helps the model escape sharp regions as it approaches convergence, without significantly increasing the training cost.

3. We conduct extensive experiments across a variety of datasets, demonstrating that ProMO consistently improves long-tailed recognition, including large-scale datasets such as Places-LT and ImageNet-LT. Our results show that ProMO effectively enhances the performance of tail classes and outperforms existing methods designed for long-tailed and class-imbalanced learning.

## 2 RELATED WORK

### 2.1 LONG-TAILED LEARNING

There have been substantial explorations in recent years to address the challenges of long-tailed learning. At the data level, re-sampling (Chawla et al., 2002; He et al., 2008) and data augmentation techniques (Zhang et al., 2018; Yun et al., 2019; Ahn et al., 2023) focus on modifying the training data distribution to mitigate class imbalance. At the representation level, decoupling frameworks (Kang et al., 2020; Xuan & Zhang, 2024) separate the feature learning stage from classifier training, allowing for independent optimization of each component. Multi-expert architectures (Wang et al., 2021b; Tan et al., 2024; Yang et al., 2024) employ multiple specialized networks to handle different class groups. Transfer learning approaches (Wang et al., 2021a; Li et al., 2024) enhance the feature space representation for minority classes with knowledge from related domains or tasks. At the loss level, re-weighting techniques (Cui et al., 2019; Luo et al., 2024) assign different weights, while margin-based techniques (Cao et al., 2019; Menon et al., 2021) impose class-specific decision boundaries during training. More recently, fine-tuning methods (Dong et al., 2023; Shi et al., 2024) adapt foundation models to long-tailed data through parameter-efficient updates that preserve generalization. Contrastive learning frameworks (Cui et al., 2021; Zhu et al., 2022; Cui et al., 2024; Du et al., 2024) have demonstrated promising results by encouraging uniformly discriminative feature representations across all classes. However, existing methods often suffer from the risk of overfitting due to limited tail class samples, highlighting the need for more robust optimization methods that can effectively navigate the complex loss landscapes inherent in imbalanced learning scenarios.

### 2.2 SHARPNESS OF LOSS LANDSCAPE

Ensuring model generalization is a fundamental yet persistent challenge in deep learning. Recent studies (Jiang et al., 2020; Stutz et al., 2021; Li et al., 2025) have empirically and theoretically demonstrated a strong connection between the geometry of the loss landscape and generalization performance, positing that models converging to flatter minima tend to generalize better than those in sharper ones. This connection becomes particularly critical in the context of imbalanced learning, where the loss landscapes associated with minority classes are often dominated by sharp regions (Zhou et al., 2023a). Traditional methods such as Perturbed Gradient Descent (Ge et al., 2015; Jin et al., 2017)

attempt to escape these regions by adding random noise to gradient updates. However, recent studies (Rangwani et al., 2022) have demonstrated that these approaches exhibit suboptimal performance in imbalanced settings, often failing to provide sufficient directional guidance to effectively navigate the complex loss landscapes of minority classes.

Sharpness-Aware Minimization (SAM) (Foret et al., 2021) has emerged as a more principled approach to address this challenge. SAM operates by identifying sharp maximal points within a neighborhood of the current parameters and subsequently minimizing the loss at these locations. Recent work has shown that SAM can be particularly effective in imbalanced settings (Rangwani et al., 2022), helping models promote convergence to flatter regions of the loss landscape. However, SAM and its variants (Zhou et al., 2023a;b; Lyu et al., 2025) require at least twice the number of gradient computations compared to standard SGD, which limits their scalability to large-scale datasets and models, where training efficiency is paramount. In this work, we aim to develop more computationally efficient methods for navigating towards flatter minima in imbalanced learning, seeking to maintain the benefits of improved optimization while reducing the associated computational burden.

## 3 METHOD

In this section, we first establish the preliminaries of our work, including the problem setup and the mechanics of the Muon optimizer. We then present our theoretical analysis of Muon, which serves as the foundation for our work by identifying its capability to escape sharp regions. Building on these findings, we introduce our primary proposed method, ProMO, a dynamic optimization method designed to leverage these theoretical benefits in a computationally efficient manner.

### 3.1 PRELIMINARIES

Let $\mathcal{D} = \{(x_i, y_i)\}_{i=1}^N$ be a training dataset of $N$ samples, where $x_i \in \mathcal{X}$ is an input sample and $y_i \in \mathcal{Y} = \{1, \ldots, C\}$ is its corresponding class label. We denote the number of samples in each class as $\{n_1, \ldots, n_C\}$, and assume, without loss of generality, that $n_i > n_j$ for any $i > j$. Real-world datasets often exhibit a long-tailed distribution with $n_1 \gg n_C$, where a small number of majority classes contain abundant samples while numerous minority classes are data-scarce. Our goal is to learn a deep neural network $h(\cdot; w)$ parameterized by $w \in \mathcal{W}$ that minimizes the empirical risk $\mathcal{L} = \frac{1}{N} \sum_{(x,y) \in \mathcal{D}} \ell(h(x; w), y)$, where $\ell$ is a loss function, such as the cross-entropy loss.

### 3.2 ANALYSIS OF MUON OPTIMIZER FROM LOSS LANDSCAPE PERSPECTIVE

To analyze the optimization dynamics, we focus our discussion, for clarity, on a single parameter matrix $\mathbf{W} \in \mathbb{R}^{m \times n}$. The principles can be extended to the entire parameter set (Kovalev, 2025). We consider two optimization methods: the standard SGD optimizer and the Muon optimizer. For the SGD optimizer, the update rule for a parameter matrix $\mathbf{W}_t$ at iteration $t$ is:

$$\mathbf{W}_{t+1} = \mathbf{W}_t - \eta_t \mathbf{g}_t, \quad \text{where} \quad \mathbf{g}_t = \nabla \mathcal{L}(\mathbf{W}_t). \tag{1}$$

Here, $\eta_t > 0$ is the learning rate and $\mathbf{g}_t$ denotes the stochastic gradient with respect to $\mathbf{W}_t$. For the Muon optimizer, the gradient is first transformed via a Newton-Schulz iteration process and then used to update the parameter $\mathbf{W}_t$. Specifically, the update is performed as:

$$\mathbf{O}_t = \text{Newton–Schulz}(\mathbf{g}_t), \quad \mathbf{W}_{t+1} = \mathbf{W}_t - \eta_t \mathbf{O}_t. \tag{2}$$

The central idea of Muon optimizer is to employ the Newton-Schulz iteration process to approximately compute the polar decomposition $\mathbf{O}_t$ of $\mathbf{g}_t$, which corresponds to $\mathbf{O}_t = \mathbf{U}_t \mathbf{V}_t^T$ in the singular value decomposition (SVD) of $\mathbf{g}_t = \mathbf{U}_t \mathbf{\Sigma}_t \mathbf{V}_t^\top$. Suppose $\mathbf{g}_t \in \mathbb{R}^{m \times n}$ is the gradient matrix with rank $r_t$, $\mathbf{\Sigma}_t \in \mathbb{R}^{r_t \times r_t}$ is a diagonal matrix containing the singular values of $\mathbf{g}_t$, $\mathbf{U}_t \in \mathbb{R}^{m \times r_t}$ and $\mathbf{V}_t \in \mathbb{R}^{n \times r_t}$ are the left and right singular vector matrices of $\mathbf{g}_t$, respectively. The update matrix becomes $\mathbf{U}_t \mathbf{V}_t^\top$, which represents the closest semi-orthogonal matrix to $\mathbf{g}_t$. Conceptually, this orthogonalization procedure maintains the structural properties of the update matrices, thereby preventing the parameters from being updated along a few dominant directions.

**Newton-Schulz Iteration Process.** This iterative process begins by normalizing the gradient matrix $\mathbf{G}_t = \mathbf{g}_t / \|\mathbf{g}_t\|_\text{F}$, where $\|\cdot\|_\text{F}$ is the Frobenius norm. The iteration is then initialized with $\mathbf{X}_0 = \mathbf{G}_t$,

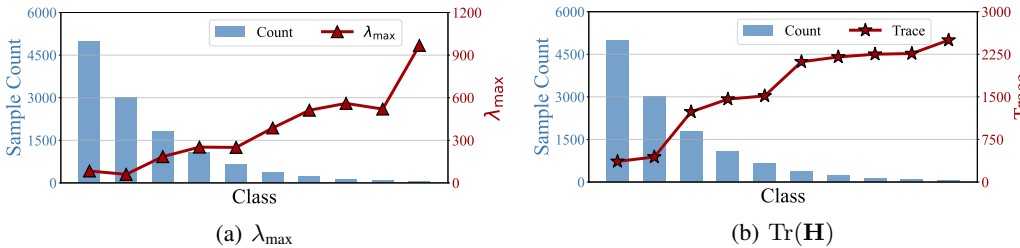

(a) $\lambda_{\max}$          (b) $\mathrm{Tr}(\mathbf{H})$

Figure 1: (a) Maximum eigenvalues $\lambda_{\max}$ ($\downarrow$) and (b) trace of Hessian metric $\mathrm{Tr}(\mathbf{H})$ ($\downarrow$) across classes with different number of samples. Classes with fewer training samples consistently exhibit larger values for both metrics, indicating that these under-represented classes converge to sharper minima in the loss landscape, which can lead to poor generalization performance.

at each step $k$ of the $N$-step iteration, $\mathbf{X}_k$ is updated from $\mathbf{X}_{k-1}$ as:

$$\mathbf{X}_k = a\mathbf{X}_{k-1} + b(\mathbf{X}_{k-1}\mathbf{X}_{k-1}^{\mathrm{T}})\mathbf{X}_{k-1} + c(\mathbf{X}_{k-1}\mathbf{X}_{k-1}^{\mathrm{T}})^2\mathbf{X}_{k-1}, \tag{3}$$

where $\mathbf{X}_N$ denotes the final output after $N$ iterative steps. The parameters $a$, $b$, and $c$ are iteration coefficients. To guarantee proper convergence of Eq. (3), these coefficients must be tuned such that the polynomial $p(x) = ax + bx^3 + cx^5$ maintains a fixed point in the neighborhood of 1. Following the original formulation (Jordan et al., 2024), we employ the coefficient values $a = 3.4445$, $b = -4.7750$, $c = 2.0315$, and perform 5 iterations. These coefficients are specifically designed to accelerate the convergence rate for matrices with small initial singular values.

**Loss Landscape in Long-Tailed Learning.** We consider the minimization of a smooth, potentially non-convex objective function $f$ (e.g. cross-entropy loss). The geometry of this landscape is often characterized by the spectral properties of its Hessian matrix $\mathbf{H}$. Key indicators of sharpness include the largest eigenvalue $\lambda_{\max}$ and the trace $\mathrm{Tr}(\mathbf{H})$, where larger value metrics indicate a sharper, more challenging optimization terrain. Following prior work (Rangwani et al., 2022), we empirically investigate this relationship by computing the eigen spectrum of the Hessian for each class on the long-tailed dataset CIFAR-10 LT. As depicted in Fig. 1, there is a clear trend where both $\lambda_{\max}$ and $\mathrm{Tr}(\mathbf{H})$ increase substantially as the number of samples per class decreases. This validates that models trained on tail classes are more prone to converging within sharper regions of the loss landscape. Consequently, an optimizer's capability to navigate towards flatter minima is paramount for achieving robust generalization, a necessity that is especially pronounced in the context of imbalanced learning.

**Escaping from Sharp Minima.** In the following, we demonstrate that the Muon algorithm can amplify the gradient projection along directions of negative curvature as training approaches convergence, thereby enabling accelerated escape from sharp areas and convergence to flatter minima. Our analysis leverages the Correlated Negative Curvature (CNC) assumption (Daneshmand et al., 2018).

**Assumption 1** (Correlated Negative Curvature (CNC)). *Let $\mathbf{W}_t$ be a point where the Hessian $\nabla^2 f(\mathbf{W}_t)$ has a minimum eigenvalue $\lambda_{min}$ at iteration $t$, and let $\mathbf{v}_{\mathbf{W}_t}$ be the corresponding unit eigenvector. The stochastic gradient $\mathbf{g}_t = \nabla f(\mathbf{W}_t)$ satisfies the CNC assumption if the second moment of its projection onto $\mathbf{v}_{\mathbf{W}_t}$ is uniformly bounded away from zero, i.e.,*

$$\exists\gamma > 0, \quad s.t. \quad \forall\mathbf{W}_t: \quad \mathbb{E}[\langle \mathbf{v}_{\mathbf{W}_t}, \mathbf{g}_t\rangle^2] \geq \gamma. \tag{4}$$

This assumption posits that the stochastic gradient has a projection along the direction of most negative curvature, providing a signal for the optimizer to move away from sharp regions. This assumption has been theoretically justified in the context of learning half-spaces and has also been empirically validated across a wide range of neural networks with varying complexity (Staib et al., 2019; Wang et al., 2020). We now present Theorem 1, with the detailed proof provided in Appendix B.

**Theorem 1.** *Let $\mathbf{W}_t$ be a point where the Hessian $\nabla^2 f(\mathbf{W}_t)$ has a minimum eigenvalue at iteration $t$, and let $\mathbf{v}_{\mathbf{W}_t}$ be the corresponding unit eigenvector. Define the projection of the SGD update onto $\mathbf{v}_{W_t}$ as $proj_{SGD} = \langle \mathbf{v}_{\mathbf{W}_t}, \mathbf{g}_t\rangle$, and the projection of the Muon update onto $\mathbf{v}_{\mathbf{W}_t}$ as $proj_{Muon} = \langle \mathbf{v}_{\mathbf{W}_t}, \mathbf{O}_t\rangle = \langle \mathbf{v}_{\mathbf{W}_t}, \mathbf{U}_t\mathbf{V}_t^{\top}\rangle$. Under the CNC assumption, the following inequality holds:*

$$\exists\gamma > 0, \quad s.t. \quad \forall\mathbf{W}_t: \quad \mathbb{E}\left[(proj_{Muon})^2\right] \geq \mathbb{E}\left[(proj_{SGD})^2\right] \geq \gamma. \tag{5}$$

**Remark.** Theorem 1 reveals that Muon's gradient orthogonalization amplifies the update's projection onto the negative curvature direction, enabling a more effective escape from sharp regions. Furthermore, prior analysis (Daneshmand et al., 2018) has shown that the convergence rate of SGD depends on the value $\gamma$ as $\mathcal{O}(\gamma^{-4})$ under certain assumptions. Our findings suggest that Muon effectively enhances the gradient component of SGD in the direction of negative curvature. Consequently, Muon is reasonably expected to converge more rapidly to the flatter minima, leading to better generalization. This aligns with the empirical evidence found in prior studies (Liu et al., 2025; Shah et al., 2025).

### 3.3 PROMO: A HYBRID OPTIMIZATION APPROACH

**Computational Overhead Analysis.** While Theorem 1 establishes the theoretical advantage of the Muon optimizer in escaping sharp regions, its computational overhead presents practical challenges for long-tailed recognition tasks. For further analysis, we estimate the FLOP overhead introduced by the Newton-Schulz iteration. For a linear layer parameterized by a weight matrix $\mathbf{W} \in \mathbb{R}^{m \times n}$, each Newton-Schulz iteration requires approximately $6mn^2$ FLOPs. For $T$ iterations, this amounts to $6Tmn^2$ FLOPs. The standard linear layer computation involves approximately $6mnL$ FLOPs, where $L$ represents the number of inputs processed (Jordan et al., 2024). For linear layers $L = B$, where $B$ denotes the batch size in tokens. Thus, the FLOP overhead $\Delta\mathcal{F}_{\text{linear}}$ for a linear layer can be estimated as:

$$\Delta\mathcal{F}_{\text{linear}} = \frac{T \cdot 6mn^2}{L \cdot 2mn} = \frac{3Tn}{B}. \tag{6}$$

For a convolutional layer, the kernel is flattened into an $m \times n$ matrix for optimization, where $m = C_{\text{out}}$ is the number of output channels and $n = C_{\text{in}} \cdot k^2$ is the product of input channels and kernel size. The number of inputs per step is $L = B \cdot H_{\text{out}} \cdot W_{\text{out}}$, where $H_{\text{out}}$ and $W_{\text{out}}$ are the spatial dimensions (height and width) of the output tensor. The FLOP overhead $\Delta\mathcal{F}_{\text{conv}}$ is then:

$$\Delta\mathcal{F}_{\text{conv}} = \frac{T \cdot 6mn^2}{L \cdot 2mn} = \frac{3TC_{\text{in}}k^2}{BH_{\text{out}}W_{\text{out}}}. \tag{7}$$

Prior studies (Jordan et al., 2024) have shown that Muon maintains FLOP overhead below 1% in large-scale language model training, where token counts per batch can reach millions (e.g., 16M tokens in LLaVA-405B). However, long-tailed recognition tasks typically employ much smaller batch sizes. This discrepancy introduces a new challenge: while Muon proves effective at escaping sharp minima, its computational overhead can become non-negligible in certain scenarios. For instance, in a ResNet layer with $C_{\text{in}} = 512, k = 3$, and an output feature map of $7 \times 7$, using $T = 5$ iterations with a batch size of $B = 256$ would result in an estimated overhead of $183\%$ according to Eq. (7). This substantial increase in training time could limit the scalability of using Muon.

**Training Dynamics Insight.** To balance computational efficiency with optimization performance, we focus on the training dynamics of SGD. Prior research (Fang et al., 2019; Rangwani et al., 2022; Abbe et al., 2023) indicates that in the early stages of training, the inherent stochasticity of SGD provides sufficient noise to effectively navigate away from sharp areas, but it behaves increasingly like deterministic gradient descent as training progresses and the learning rate decays, making it more prone to stalling near sharp regions late in training, especially in long-tailed scenarios. This observation suggests that the capability of Muon to reach flatter minima is most valuable during later training phases when SGD's inherent noise becomes insufficient.

**Dynamic Hybrid Optimization.** We propose ProMO, a hybrid optimization method that dynamically alternates between SGD and Muon. Specifically, for a training process with $T_{\max}$ total epochs, at epoch $t$, the model applies a Muon update with probability $p_t \in [0, 1]$ and otherwise applies a SGD update with probability $1 - p_t$. We define $p_t$ using a sinusoidal schedule:

$$p_t = \sin\left(\frac{\pi}{2} \cdot \frac{t}{T_{\max}}\right). \tag{8}$$

This sinusoidal probability schedule ensures that during early training, SGD is predominantly selected (*i.e.,* Eq. (1)), leveraging its inherent stochasticity for escaping sharp regions while minimizing Muon's computational overhead. As training progresses, the probability of selecting Muon gradually increases (*i.e.,* Eq. (2)), providing enhanced capabilities to escape sharp minima that may hinder generalization performance when SGD's noise becomes insufficient. This dynamic approach maximizes Muon's benefits while maintaining computational efficiency. Notably, both optimizers operate on the same

parameter set, with Muon simply applying orthogonalization to SGD's gradient updates without introducing additional state or parameters to the optimization process. The pseudo-code for the training processes of Muon and our ProMO are provided in Appendix A.

# 4 EXPERIMENTS

## 4.1 EXPERIMENTAL SETUP

**Datasets.** We evaluate the proposed Muon optimizer method on a suite of widely used long-tailed benchmarks, CIFAR-10 LT, CIFAR-100 LT, ImageNet-LT (Liu et al., 2019), and Places-LT (Liu et al., 2019). CIFAR-10 LT and CIFAR-100 LT are two long-tailed datasets sampled from the original CIFAR datasets (Krizhevsky et al., 2009), consisting of 10 and 100 classes, respectively. We conduct experiments under varying imbalance factors, defined as IF $= n_{\max}/n_{\min}$, where $n_{\max}$ and $n_{\min}$ denote the number of samples in the most and least frequent classes, respectively. Following the mainstream protocol (Cui et al., 2019), we adopt imbalance settings with imbalance factors of 10 and 100, where the number of samples per class decreases exponentially. ImageNet-LT is a large-scale long-tailed dataset derived from the ImageNet dataset (Deng et al., 2009), comprising 115.8k training images across 1,000 categories, with class frequencies ranging from 1,280 to 5 instances, and an imbalance factor of 256. Places-LT contains 62.5k training images from 365 scene categories, with the number of samples per class varying from 4,980 to 5, and an imbalance factor of 996.

**Evaluation Protocol.** We follow standard protocols (Wang et al., 2023) in long-tailed classification by treating all classes equally during testing and reporting results across three class splits: *Many*, *Medium*, and *Few*, based on the number of training samples per class. Consistent with prior work (Cui et al., 2019; Rangwani et al., 2022), we use top-1 accuracy as our evaluation metric and report it for each class split as well as overall on each dataset. To assess computational efficiency, we also record the average training time per epoch associated with each method across all datasets.

**Baselines.** We compare our method with a range of strong baselines commonly used in long-tailed classification. We evaluate four optimizers: SGD, SAM, Muon, and our proposed ProMO, applied to various widely used methods: Cross-Entropy (CE), Class-Balanced Loss (CB) (Cui et al., 2019), Logit Adjustment (LA) (Menon et al., 2021), Balanced Contrastive Learning (BCL) (Zhu et al., 2022), and Probabilistic Contrastive Learning (ProCo) (Du et al., 2024). This allows us to comprehensively assess the contribution of our optimizer across various long-tailed learning paradigms.

**Implementation details.** Our code is implemented with Pytorch 1.12.1. All experiments are carried out on NVIDIA GeForce RTX 3090 GPUs. For a fair comparison, we use ResNet32 on CIFAR-10 LT and CIFAR-100 LT, ResNet50 on ImageNet-LT, and pre-trained ResNet-152 on Places-LT. We train each model using a batch size of 256 (for CIFAR-10 LT and CIFAR-100 LT) / 128 (for ImageNet-LT) / 512 (for Places-LT), with a momentum of 0.9 and a weight decay of 0.0002. We adopt the Nesterov momentum form for all optimizers, with an initial learning rate of 0.1; a multi-step schedule (decayed to 0.01 and 0.0001 at epochs 160 and 180) for CIFAR-10 LT and CIFAR-100 LT, and a cosine schedule throughout training for ImageNet-LT and Places-LT. For Newton-Schulz iteration steps $N$ in the Muon optimizer, we set $N = 5$ for the sake of efficiency.

## 4.2 COMPARISON RESULTS

**Results on CIFAR-10 LT and CIFAR-100 LT.** We first evaluate Muon and ProMO on CIFAR-10 LT and CIFAR-100 LT under imbalance factors (IF) of 10 and 100. As shown in Table 1, both methods consistently outperform SGD and SAM across all class subsets (Many, Medium, Tail), with the largest gains on tail classes under severe imbalance. On CIFAR-10 LT, Muon achieves clear improvements over both baselines, particularly at IF=100 where it boosts tail accuracy without compromising head or medium classes. The advantage is even more pronounced on CIFAR-100 LT: Muon not only improves overall accuracy under both moderate and extreme imbalance, but also delivers substantial gains for tail classes; for instance, when paired with the CB loss, Muon improves tail class accuracy by 2.2% (IF=10) and 2.4% (IF=100) over SGD. Across all conditions, Muon maintains a consistent edge over SAM, highlighting its robustness in long-tailed learning.

Crucially, our proposed ProMO not only matches but in some cases even surpasses the performance of the Muon optimizer across various experimental settings. This demonstrates that our dynamic

Table 1: Top-1 accuracy (%) (↑) results for *Many*, *Medium* (namely Med.), *Few* and overall classes on CIFAR-10 LT and CIFAR-100 LT datasets, categorized by imbalance factors (IF) of 10 and 100. ProMO and Muon are highlighted in blue to group them for focused comparison against the baselines.

| Loss | Method | CIFAR-10 LT IF = 100 | | | | CIFAR-100 LT IF = 10 | | | | CIFAR-100 LT IF = 100 | | | |
|---|---|---|---|---|---|---|---|---|---|---|---|---|---|
| | | Many | Med. | Few | All | Many | Med. | Few | All | Many | Med. | Few | All |
| CE | SGD | 94.1 | 77.4 | 65.0 | 77.4 | 75.6 | 62.8 | 48.2 | 60.8 | 75.9 | 52.0 | 15.7 | 44.6 |
| | SAM | 95.7 | 76.7 | 64.4 | 77.5 | 76.7 | 64.4 | 49.0 | 61.9 | 77.5 | 51.1 | 15.8 | 44.9 |
| | **ProMO** | 95.2 | 76.9 | 64.3 | 77.3 | 77.1 | 65.4 | 49.7 | **62.6** | 77.2 | 53.9 | 16.2 | **45.8** |
| | **Muon** | 95.1 | 75.7 | 67.1 | **78.1** | 75.0 | 65.6 | 49.1 | 61.8 | 77.2 | 52.4 | 17.3 | **45.8** |
| CB | SGD | 94.8 | 77.0 | 66.0 | 77.9 | 75.1 | 63.5 | 48.4 | 60.9 | 75.0 | 50.6 | 17.3 | 44.6 |
| | SAM | 94.9 | 76.0 | 65.8 | 77.6 | 77.5 | 65.1 | 48.3 | 62.1 | 75.4 | 50.6 | 19.0 | 45.4 |
| | **ProMO** | 94.9 | 77.1 | 66.7 | **78.3** | 76.1 | 66.4 | 50.5 | **62.9** | 76.5 | 52.5 | 19.3 | 46.4 |
| | **Muon** | 95.1 | 77.9 | 66.0 | **78.3** | 76.4 | 65.9 | 50.6 | **62.9** | 76.4 | 52.2 | 19.7 | **46.5** |
| LA | SGD | 90.3 | 76.9 | 80.9 | 82.5 | 70.0 | 64.3 | 57.1 | 63.2 | 69.2 | 53.6 | 34.3 | 50.5 |
| | SAM | 91.9 | 78.2 | 81.9 | 83.8 | 72.6 | 64.5 | 58.8 | 64.6 | 67.6 | 54.6 | 35.8 | 51.0 |
| | **ProMO** | 91.5 | 78.0 | 82.0 | 83.7 | 71.9 | 65.0 | 58.6 | 64.5 | 69.3 | 55.0 | 34.8 | 51.2 |
| | **Muon** | 92.6 | 79.9 | 82.4 | **84.7** | 71.7 | 65.0 | 59.3 | **64.7** | 68.5 | 56.2 | 36.1 | **51.9** |
| BCL | SGD | 93.2 | 79.3 | 81.7 | 84.4 | 71.7 | 64.5 | 59.5 | 64.7 | 68.5 | 54.2 | 34.2 | 50.5 |
| | SAM | 94.0 | 80.8 | 82.7 | 85.5 | 72.5 | 65.2 | 60.0 | 65.3 | 68.1 | 53.5 | 37.1 | 51.3 |
| | **ProMO** | 93.9 | 80.2 | 82.1 | 85.1 | 73.9 | 66.0 | 60.4 | **66.1** | 71.1 | 57.5 | 36.3 | **53.1** |
| | **Muon** | 94.3 | 80.5 | 82.9 | **85.6** | 73.2 | 66.3 | 60.5 | 66.0 | 70.7 | 56.4 | 36.9 | 52.9 |
| ProCo | SGD | 93.6 | 80.7 | 82.2 | 85.2 | 71.8 | 64.7 | 59.2 | 64.6 | 68.9 | 55.4 | 36.2 | 51.8 |
| | SAM | 92.6 | 80.3 | 84.7 | 85.8 | 73.6 | 64.2 | 59.9 | 65.3 | 69.4 | 56.2 | 36.7 | 52.4 |
| | **ProMO** | 94.2 | 81.0 | 83.3 | **85.9** | 73.6 | 67.3 | 59.6 | 66.1 | 70.1 | 57.1 | 36.9 | 52.9 |
| | **Muon** | 94.2 | 81.6 | 82.9 | **85.9** | 73.8 | 65.4 | 61.5 | **66.4** | 70.0 | 57.4 | 37.2 | **53.1** |

Table 2: Top-1 accuracy (%) (↑) results for *Many*, *Medium* (namely Med.), *Few*, and overall classes on ImageNet-LT (IN-LT) and Places-LT (PL-LT) datasets, categorized by different loss functions. ProMO and Muon are highlighted in blue to group them for focused comparison against the baselines.

| Dataset | Method | CE | | | | LA | | | | ProCo | | | |
|---|---|---|---|---|---|---|---|---|---|---|---|---|---|
| | | Many | Med. | Few | All | Many | Med. | Few | All | Many | Med. | Few | All |
| IN-LT | SGD | 69.4 | 42.2 | 14.8 | 49.0 | 64.3 | 52.4 | 35.1 | 54.6 | 66.3 | 54.3 | 37.8 | 56.7 |
| | SAM | 71.7 | 43.7 | 16.1 | 50.7 | 66.1 | 54.5 | 38.5 | 56.8 | 66.8 | 56.9 | 40.2 | 58.5 |
| | **ProMO** | 72.7 | 45.1 | 16.2 | **51.8** | 67.4 | 54.2 | 37.8 | 57.1 | 68.4 | 56.6 | 41.1 | **59.0** |
| | **Muon** | 72.5 | 44.1 | 16.1 | 51.2 | 68.5 | 54.5 | 37.4 | **57.6** | 67.3 | 56.0 | 39.5 | 58.1 |
| PL-LT | SGD | 46.3 | 22.0 | 4.4 | 27.3 | 42.0 | 40.3 | 27.4 | 38.4 | 43.6 | 42.0 | 26.4 | 39.5 |
| | SAM | 47.0 | 25.2 | 9.1 | 29.9 | 42.1 | 42.2 | 33.3 | 40.4 | 42.9 | 42.6 | 30.3 | 40.3 |
| | **ProMO** | 47.0 | 25.2 | 9.1 | 29.9 | 43.3 | 41.7 | 32.5 | **40.5** | 43.4 | 42.1 | 33.0 | **40.8** |
| | **Muon** | 47.6 | 26.9 | 10.7 | **31.2** | 43.4 | 41.6 | 33.1 | **40.5** | 43.4 | 42.2 | 31.9 | 40.6 |

optimization strategy successfully captures the benefits of Muon's gradient orthogonalization during critical training phases while maintaining computational efficiency, as will be detailed in Table 4. Additionally, as shown in Fig. 3(c), experiments on both the balanced and imbalanced versions of CIFAR-100 demonstrate that Muon is particularly effective in enhancing generalization performance under imbalanced settings. See Appendix C.1 for more comparison results.

**Results on Large-Scale Datasets.** The benefits of Muon become more pronounced on large-scale datasets which present far more extreme class imbalance and substantially larger numbers of classes. As detailed in Table 2, Muon delivers consistent and notable gains, particularly for the under-represented medium and tail classes. On Places-LT, Muon significantly improves overall accuracy over SGD by 1.1% to 3.9% across various loss functions. Critically, its impact is most profound on the tail classes, boosting their accuracy by up to a remarkable 6.3% (with CE loss). Furthermore, Muon consistently achieves superior or competitive performance compared to SAM, demonstrating its ability to find superior generalizing solutions. These trends hold on ImageNet-LT. Muon again surpasses SGD, with overall accuracy improving by 1.4% to 3.0%. The benefit for tail classes remains significant, confirming the robustness of our method under diverse, challenging conditions.

Table 3: Loss landscape geometry metrics for SGD and Muon on CIFAR-10 LT and CIFAR-100 LT with imbalance factor 100. Maximum eigenvalue $\lambda_{\max}$ ($\downarrow$) and trace of Hessian matrix $\text{Tr}(\mathbf{H})$ ($\downarrow$) are reported for the three least frequent classes. Superscripts (1), (2), and (3) denote the 1st, 2nd, and 3rd rarest classes, respectively. Lower values indicate flatter minima and improved generalization. Performance of Muon is highlighted in blue for focused comparison.

| Dataset | Method | $\lambda_{\max}^{(1)}$ | $\lambda_{\max}^{(2)}$ | $\lambda_{\max}^{(3)}$ | $\text{Tr}(\mathbf{H})^{(1)}$ | $\text{Tr}(\mathbf{H})^{(2)}$ | $\text{Tr}(\mathbf{H})^{(3)}$ |
|---|---|---|---|---|---|---|---|
| CIFAR-10 LT | SGD | 968.51 | 516.88 | 560.69 | 2499.73 | 2263.09 | 2251.20 |
|  | **Muon** | **116.02** | **115.23** | **95.00** | **404.58** | **377.77** | **412.38** |
| CIFAR-100 LT | SGD | 929.56 | 524.80 | 404.30 | 1560.40 | 1604.30 | 1843.20 |
|  | **Muon** | **358.20** | **309.71** | **322.10** | **437.58** | **529.98** | **810.12** |

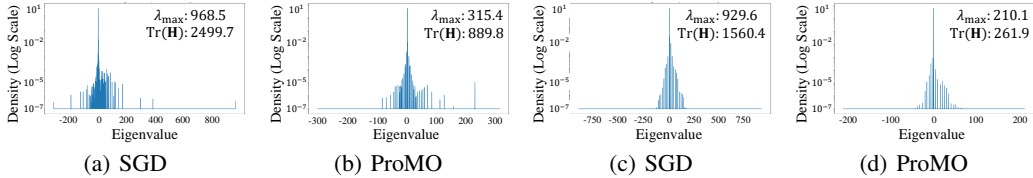

(a) SGD      (b) ProMO      (c) SGD      (d) ProMO

Figure 2: Eigen spectral density for the class with the fewest samples across different methods. Experiments are conducted on (a,b) CIFAR-10 LT and (c,d) CIFAR-100 LT with imbalance factor 100. Maximum eigenvalue $\lambda_{\max}$ ($\downarrow$) and trace of hessian metric $\text{Tr}(\mathbf{H})$ ($\downarrow$) in the top right corner of each panel. Lower $\lambda_{\max}$ and $\text{Tr}(\mathbf{H})$ indicate a smoother loss landscape and improved generalization.

Notably, our proposed ProMO continues to exhibit strong performance on large-scale benchmarks, demonstrating a clear advantage over both SGD and SAM. Its accuracy is largely on par with the Muon optimizer. Intriguingly, when combined with the most effective loss function, ProCo, ProMO not only matches but surpasses the performance of the Muon optimizer on both datasets. This suggests that the dynamic scheduling of optimizers may introduce a more diverse optimization pathway, potentially guiding the model towards wider, better-generalizing minima than either optimizer could find alone.

**Flat Minima of Loss Landscape.** To further investigate the mechanism behind the improved generalization performance of tail classes, we analyze the optimization from a loss landscape perspective. We compute the eigenvalue spectrum of the Hessian matrix for tail classes on both CIFAR-10 LT and CIFAR-100 LT datasets with an imbalance factor of 100, training with the LA loss, as shown in Table 3 and Fig. 2. Table 3 presents the Hessian properties for the three classes with the smallest sample sizes, comparing SGD and Muon optimizers through the maximum eigenvalue $\lambda_{\max}$ and trace $\text{Tr}(\mathbf{H})$ at convergence. Smaller values indicate flatter loss landscapes associated with better generalization. The results show clear advantages of Muon: on CIFAR-10 LT, $\lambda_{\max}$ drops by 77%–88% and the trace by 81%–84% relative to SGD. On the more challenging CIFAR-100 LT, reductions remain substantial, with 20%–61% in $\lambda_{\max}$ and 56%–72% in the trace. These findings indicate that Muon drives tail classes toward flatter minima, consistent with our theoretical analysis.

It is also crucial to examine the optimizer's ability to escape saddle points, as the loss landscape of tail classes often exhibits highly negative minimum eigenvalues, indicating convergence to such regions. To validate this, we compute the minimum eigenvalues ($\lambda_{min}$) for the class with the fewest samples under Muon and SGD on both CIFAR-10 LT and CIFAR-100 LT under IF=100. On CIFAR-10 LT, the $\lambda_{min}$ under Muon is -110.33, which is substantially larger than the -316.30 observed under SGD. This trend is even more pronounced on the challenging CIFAR-100 LT dataset, where Muon achieved a $\lambda_{min}$ of -352.22 compared to -916.12 for SGD. This observation indicates significantly weaker negative curvature, corroborating Muon's effectiveness in escaping saddle-like regions for tail classes.

**Computational Efficiency Analysis.** We analyze the computational efficiency of ProMO against the SGD, SAM, and Muon optimizers, as shown in Table 4, Figs. 3(a) and 3(b). In Table 4, we measure the average training time per epoch across four datasets, using two representative loss functions, LA and ProCo, to evaluate the robustness of each optimizer to varying loss complexities.

The results demonstrate that ProMO effectively resolves the trade-off between generalization and computational cost, achieving the strong performance of Muon with minimal overhead. This efficiency

Table 4: Computational overhead of different optimizers on long-tailed benchmarks. We report the average training time per epoch (seconds) (↓) and the runtime ratio relative to SGD (in parentheses). Muon and ProMO are highlighted in blue to group them for focused comparison against the baselines.

| Loss | Method | CIFAR-100 LT | | | | ImageNet-LT | | Places-LT | |
|---|---|---|---|---|---|---|---|---|---|
| | | IF=10 | | IF=100 | | | | | |
| LA | SGD | 4.2s | (1.00×) | 3.5s | (1.00×) | 184.8s | (1.00×) | 174.0s | (1.00×) |
| | SAM | 7.0s | (1.67×) | 4.4s | (1.27×) | 392.2s | (2.21×) | 216.0s | (1.24×) |
| | **Muon** | 8.4s | (1.99×) | 5.5s | (1.61×) | 358.8s | (1.94×) | 472.8s | (2.71×) |
| | **ProMO** | 5.9s | (1.41×) | 4.2s | (1.23×) | 244.2s | (1.32×) | 204.0s | (1.17×) |
| ProCo | SGD | 11.1s | (1.00×) | 7.0s | (1.00×) | 684.0s | (1.00×) | 622.8s | (1.00×) |
| | SAM | 21.3s | (1.92×) | 13.0s | (1.85×) | 1870.2s | (2.73×) | 2829.0s | (4.54×) |
| | **Muon** | 17.6s | (1.59×) | 11.3s | (1.60×) | 1118.4s | (1.63×) | 1608.0s | (2.58×) |
| | **ProMO** | 12.4s | (1.12×) | 8.2s | (1.17×) | 804.0s | (1.17×) | 874.8s | (1.40×) |

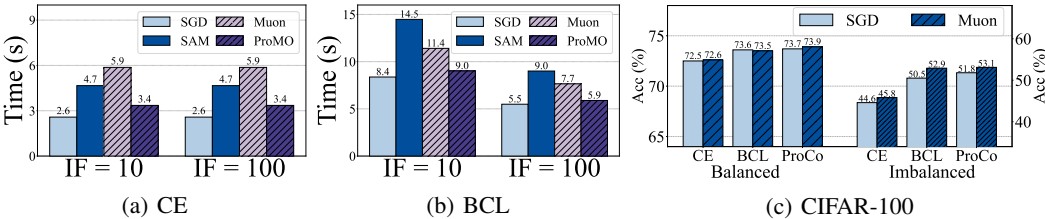

(a) CE  (b) BCL  (c) CIFAR-100

Figure 3: (a, b) Average training time per epoch (seconds) (↓) for various methods on CIFAR-100 LT with imbalance factors of 10 and 100, using (a) LA and (b) BCL loss functions, respectively. (c) Top-1 accuracy (%) (↑) comparison between SGD and Muon on both the standard balanced CIFAR-100 and its long-tailed version (IF=100) across various loss functions. Results demonstrate that Muon is particularly effective for improving generalization performance in imbalanced settings.

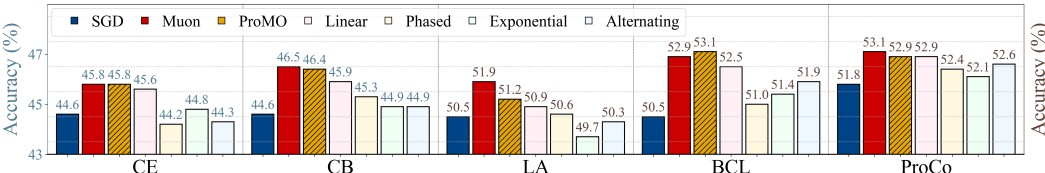

Figure 4: Top-1 accuracy (%) (↑) comparison of the proposed sinusoidal probability scheduling method in ProMO against four alternative probability schedules across different loss functions. Experiments are conducted on CIFAR-100 LT with an imbalance factor of 100.

is most striking with complex losses like ProCo, where ProMO adds only 21% training overhead compared to SGD. In sharp contrast, SAM incurs a 176% overhead and Muon still bears a considerable 85%, indicating their pronounced scalability limitations when integrated with advanced long-tailed learning methods. When paired with LA loss, ProMO maintains its advantage, incurring just 25% overhead versus 36% for SAM and a costly 106% for Muon. Importantly, these dramatic efficiency gains come at no cost to accuracy. As shown in Tables 1 and 2, ProMO consistently matches the performance of the Muon optimizer, establishing it as a highly practical and scalable method for real-world, large-scale long-tail recognition. See Appendix C.2 for more comparison results.

**Comparison with Alternative Probability Schedules.** To further assess the effectiveness of our sinusoidal probability scheduling method in Eq. (8), we compared it against several alternative scheduling approaches. Specifically, we evaluated: (1) *Linear*, where the probability of selecting Muon increases linearly from 0 to 1; (2) *Phased*, which employs SGD exclusively in the first half of training and switches entirely to Muon in the second half; (3) *Exponential*, where the probability of choosing Muon grows exponentially from 0 to 1; and (4) *Alternating*, where the optimizer alternates between Muon and SGD at each epoch. The experimental results on CIFAR-100 LT with an imbalance factor of 100 are presented in Fig. 4. While most of these schedules generally outperform pure SGD

in terms of accuracy, the sinusoidal probability schedule consistently achieves superior performance across a variety of loss functions. This finding highlights that gradually biasing the training process toward Muon in a sinusoidal manner offers more stability and adaptability, ultimately enabling stronger generalization compared to other probability scheduling methods.

## 5 CONCLUSION

In this work, we present a theoretical analysis of the Muon optimizer from the perspective of loss landscape geometry and introduce ProMO, a novel hybrid optimization approach designed to address the poor generalization of tail classes in long-tailed recognition. Our approach is grounded in new insight demonstrating that the Muon optimizer effectively escapes sharp minima by enhancing the gradient's projection along directions of negative curvature. To mitigate Muon's computational overhead, ProMO dynamically chooses between standard SGD and Muon optimization using a sinusoidal schedule that progressively favors Muon as training converges. This approach strikes an effective balance between computational efficiency and performance, guiding the model toward flatter loss landscapes and significantly improving generalization on tail classes, as validated by extensive experiments. For future work, we will investigate the efficacy of our approach in other imbalanced learning scenarios, such as domain adaptation, to further enhance its applicability and robustness.

## ETHICS STATEMENT

This work complies with the Code of Ethics. It uses only publicly available datasets, involves no human or sensitive data, and raises no foreseeable risks related to privacy, security, or fairness issues. The research is conducted solely for scientific advancement, with no conflicts of interest.

## REPRODUCIBILITY STATEMENT

We are committed to ensure the reproducibility of our proposed method. A detailed description of our approach is provided in Section 3.3, and the corresponding source code will be made publicly available upon publication of this paper. Both backbone models and datasets used in our work are publicly available. Furthermore, the detailed experimental settings required for reproducing are presented in Section 4.1. We believe that these components provide the community with details necessary to verify and build upon our work.

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

## A  ALGORITHM

We present the pseudo-code of Muon and ProMO in Algorithm 1 and Algorithm 2, respectively, to illustrate the detailed implementation procedure of our method.

---

**Algorithm 1** Muon

---

**Input**: Initial weights $\mathbf{W}_0$, learning rate schedule $\{\eta_t\}$, momentum $\beta$, batch size $B$, dataset $\mathcal{D}$
**for** $t = 0$ to $T_{\max} - 1$ **do**
    Sample mini batch $\{\xi_{t,i}\}_{i=1}^{B} \leftarrow \mathcal{D}$
    Calculate $\mathbf{g}_t = \frac{1}{B} \sum_{i=1}^{B} \nabla f(\mathbf{W}_t; \xi_{t,i})$
    If $t > 0$, $\mathbf{M}_t = \beta \mathbf{M}_{t-1} + (1-\beta)\mathbf{g}_t$. If $t = 0$, $\mathbf{M}_0 = \mathbf{g}_0$
    Calculate $\mathbf{O}_t = \text{NewtonSchulz}(\mathbf{M}_t)$
    Update $\mathbf{W}_{t+1} = \mathbf{W}_t - \eta_t \mathbf{O}_t$
**end for**

---

---

**Algorithm 2** ProMO

---

**Input**: Initial weights $\mathbf{W}_0$, learning rate schedule $\{\eta_t\}$, momentum $\beta$, batch size $B$, dataset $\mathcal{D}$
**for** $t = 0$ to $T_{\max} - 1$ **do**
    Sample mini batch $\{\xi_{t,i}\}_{i=1}^{B} \leftarrow \mathcal{D}$
    Calculate $\mathbf{g}_t = \frac{1}{B} \sum_{i=1}^{B} \nabla f(\mathbf{W}_t; \xi_{t,i})$
    If $t > 0$, $\mathbf{M}_t = \beta \mathbf{M}_{t-1} + (1-\beta)\mathbf{g}_t$. If $t = 0$, $\mathbf{M}_0 = \mathbf{g}_0$
    Calculate $p_t$ via Eq. (8)
    Sample $\mu \sim \text{Uniform}(0, 1)$
    If $\mu < p_t$ then $\mathbf{O}_t = \text{NewtonSchulz}(\mathbf{M}_t)$ else $\mathbf{O}_t = \mathbf{M}_t$
    Update $\mathbf{W}_{t+1} = \mathbf{W}_t - \eta_t \mathbf{O}_t$
**end for**

---

## B  THEORETICAL SUPPLEMENT

**Lemma 1.** *Given the normalized gradient matrix* $\mathbf{G}_t = \frac{\mathbf{g}_t}{\|\mathbf{g}_t\|_F}$ *and its rank-$r_t$ singular value decomposition* $\mathbf{G}_t = \mathbf{U}_t \mathbf{S}_t \mathbf{V}_t^\top$, *where* $\mathbf{U}_t \in \mathbb{R}^{m \times r_t}$ *and* $\mathbf{V}_t \in \mathbb{R}^{n \times r_t}$ *satisfy* $\mathbf{U}_t^\top \mathbf{U}_t = \mathbf{I}_{r_t}$ *and* $\mathbf{V}_t^\top \mathbf{V}_t = \mathbf{I}_{r_t}$, *and* $\mathbf{S}_t = \text{diag}(s_1, \ldots, s_{r_t}) \in \mathbb{R}^{r_t \times r_t}$ *is the diagonal matrix of singular values, it holds that* $\sum_{i=1}^{r_t} s_i^2 = 1$.

*Proof of Lemma 1.* By the normalization condition $\mathbf{G}_t = \mathbf{g}_t / \|\mathbf{g}_t\|_F$, the Frobenius norm of $\mathbf{G}_t$ is:

$$\|\mathbf{G}_t\|_F = \frac{\|\mathbf{g}_t\|_F}{\|\mathbf{g}_t\|_F} = 1, \tag{9}$$

which implies $\|\mathbf{G}_t\|_F^2 = 1$. The squared Frobenius norm is equivalent to the trace of $\mathbf{G}_t^\top \mathbf{G}_t$:

$$\|\mathbf{G}_t\|_F^2 = \text{trace}(\mathbf{G}_t^\top \mathbf{G}_t). \tag{10}$$

Substituting the SVD, $\mathbf{G}_t = \mathbf{U}_t \mathbf{S}_t \mathbf{V}_t^\top$, we compute:

$$\mathbf{G}_t^\top \mathbf{G}_t = (\mathbf{U}_t \mathbf{S}_t \mathbf{V}_t^\top)^\top (\mathbf{U}_t \mathbf{S}_t \mathbf{V}_t^\top) = \mathbf{V}_t \mathbf{S}_t^\top \mathbf{U}_t^\top \mathbf{U}_t \mathbf{S}_t \mathbf{V}_t^\top = \mathbf{V}_t \mathbf{S}_t^\top \mathbf{I}_{r_t} \mathbf{S}_t \mathbf{V}_t^\top = \mathbf{V}_t \mathbf{S}_t^2 \mathbf{V}_t^\top. \tag{11}$$

The trace operation yields:

$$\|\mathbf{G}_t\|_F^2 = \text{tr}(\mathbf{V}_t \mathbf{S}_t^2 \mathbf{V}_t^\top) = \text{tr}(\mathbf{V}_t^\top \mathbf{V}_t \mathbf{S}_t^2) = \text{tr}(\mathbf{I}_{r_t} \cdot \mathbf{S}_t^2) = \text{tr}(\mathbf{S}_t^2). \tag{12}$$

The matrix $\mathbf{S}_t^2 = \text{diag}(s_1^2, \ldots, s_{r_t}^2)$ is diagonal, so its trace is the sum of the squared singular values:

$$\text{tr}(\mathbf{S}_t^2) = \sum_{i=1}^{r_t} s_i^2. \tag{13}$$

Combining these results, we conclude:

$$\sum_{i=1}^{r_t} s_i^2 = \|\mathbf{G}_t\|_{\mathrm{F}}^2 = 1. \tag{14}$$

**Lemma 2.** *Let matrices $\mathbf{A}, \mathbf{B} \in \mathbb{R}^{r_t \times r_t}$ be positive semi-definite (PSD) matrices. Then, it holds that $\mathrm{tr}(\mathbf{AB}) \geq 0$.*

*Proof of Lemma 2.* Since $\mathbf{A}$ is PSD, it admits a symmetric PSD square root $\mathbf{A}^{1/2}$ satisfying $\mathbf{A} = \mathbf{A}^{1/2}\mathbf{A}^{1/2}$ and $(\mathbf{A}^{1/2})^\top = \mathbf{A}^{1/2}$. Applying the cyclic property of the trace operator, we reinterpret $\mathrm{tr}(\mathbf{AB})$ as:

$$\mathrm{tr}(\mathbf{AB}) = \mathrm{tr}\left(\mathbf{A}^{1/2}(\mathbf{A}^{1/2}\mathbf{B})\right) = \mathrm{tr}\left(\mathbf{A}^{1/2}\mathbf{B}\mathbf{A}^{1/2}\right). \tag{15}$$

The matrix $\mathbf{M} = \mathbf{A}^{1/2}\mathbf{B}\mathbf{A}^{1/2}$ preserves the PSD property: for any vector $\mathbf{x} \in \mathbb{R}^{r_t}$,

$$\mathbf{x}^\top \mathbf{M}\mathbf{x} = \left(\mathbf{x}^\top \mathbf{A}^{1/2}\right)\mathbf{B}\left((\mathbf{A}^{1/2})^\top \mathbf{x}\right) = \mathbf{y}^\top \mathbf{B}\mathbf{y} \geq 0, \quad \mathbf{y} = \mathbf{A}^{1/2}\mathbf{x}, \tag{16}$$

since $\mathbf{B}$ is PSD. Consequently, $\mathbf{M}$ is also PSD, and its trace—equivalent to the sum of its non-negative eigenvalues—satisfies $\mathrm{tr}(\mathbf{M}) \geq 0$, that is,

$$\mathrm{tr}(\mathbf{AB}) = \mathrm{tr}\left(\mathbf{A}^{1/2}\mathbf{B}\mathbf{A}^{1/2}\right) \geq 0. \tag{17}$$

*Proof of Theorem 1.* Define $\mathbf{V} \in \mathbb{R}^{m \times n}$ as the matrix obtained by reshaping $\mathbf{v}_{\mathbf{W}_t}$ into shape $m \times n$, so that $\mathrm{vec}(\mathbf{V}) = \mathbf{v}_{\mathbf{W}_t}$. The SGD update direction is the stochastic gradient $\mathbf{g}_t$, and its projection onto $\mathbf{v}_{\mathbf{W}_t}$ is:

$$\mathrm{proj}_{\mathrm{SGD}} = \mathrm{vec}(\mathbf{g}_t)^\top \mathrm{vec}(\mathbf{V}) = \mathrm{tr}(\mathbf{g}_t^\top \mathbf{V}). \tag{18}$$

Remember that $\mathbf{G}_t = \mathbf{g}_t / \|\mathbf{g}_t\|_{\mathrm{F}}$ is the normalized gradient. The projection of $\mathbf{G}_t$ onto $\mathbf{v}_{\mathbf{W}_t}$ can be expressed as:

$$\mathrm{tr}(\mathbf{G}_t^\top \mathbf{V}) = \mathrm{tr}(\mathbf{V}_t \mathbf{S}_t \mathbf{U}_t^\top \mathbf{V}) = \mathrm{tr}(\mathbf{S}_t \mathbf{M}) = \sum_{i=1}^{r_t} s_i\, m_{ii}, \tag{19}$$

where $\mathbf{M}_t := \mathbf{U}_t^\top \mathbf{V}\mathbf{V}_t \in \mathbb{R}^{r_t \times r_t}$, and $s_i \in [0,1]$ are the singular values of $\mathbf{G}_t$, while $m_{ii}$ are the diagonal entries of $\mathbf{M}$. Hence, the SGD projection on $\mathbf{v}_{\mathbf{W}_t}$ becomes:

$$\mathrm{proj}_{\mathrm{SGD}} = \|\mathbf{g}_t\|_{\mathrm{F}} \cdot \mathrm{tr}(\mathbf{G}_t^\top \mathbf{V}) = \|\mathbf{g}_t\|_{\mathrm{F}} \cdot \mathrm{tr}(\mathbf{S}_t \mathbf{M}). \tag{20}$$

The Muon update direction is given by $\mathbf{U}_t \mathbf{V}_t^\top$, and its projection onto $\mathbf{v}_{\mathbf{W}_t}$ is:

$$\begin{aligned}
\mathrm{proj}_{\mathrm{Muon}} &= \mathrm{vec}(\mathbf{U}_t \mathbf{V}_t^\top)^\top \mathrm{vec}(\mathbf{V}) = \mathrm{tr}((\mathbf{U}_t \mathbf{V}_t^\top)^\top \mathbf{V}) \\
&= \mathrm{tr}(\mathbf{V}_t \mathbf{U}_t^\top \mathbf{V}) = \mathrm{tr}(\mathbf{U}_t^\top \mathbf{V}\mathbf{V}_t) = \mathrm{tr}(\mathbf{M}) = \sum_{i=1}^{r_t} m_{ii}.
\end{aligned} \tag{21}$$

Now we compare the expected squared projection of Muon and the normalized gradient $\mathbf{G}_t$ onto $\mathbf{v}_{\mathbf{W}_t}$.

$$\begin{aligned}
\mathbb{E}\big[(\mathrm{tr}(\mathbf{M}_t))^2\big] - \mathbb{E}\big[(\mathrm{tr}(\mathbf{S}_t \mathbf{M}_t)^2\big] &= \mathbb{E}\left[\left(\sum_i m_{ii}\right)^2 - \left(\sum_i s_i m_{ii}\right)^2\right] \\
&= \mathbb{E}\left[\sum_{i,j}(1 - s_i s_j)m_{ii}m_{jj}\right].
\end{aligned} \tag{22}$$

Let us define vector $\mathbf{s} = [s_1, \ldots, s_{r_t}]^\top \in \mathbb{R}^{r_t}$, where $\|\mathbf{s}\|_2 = 1$ since $\mathbf{G}_t$ is normalized. Then, define the matrix $\mathbf{A} := \mathbf{I} - \mathbf{s}\mathbf{s}^\top \in \mathbb{R}^{r_t \times r_t}$. Let vector $\mathbf{m} = [m_{11}, \ldots, m_{r_t r_t}]^\top \in \mathbb{R}^{r_t}$, and define the matrix $\mathbf{B} := \mathbf{m}\mathbf{m}^\top \in \mathbb{R}^{r_t \times r_t}$. We now show that both matrices $\mathbf{A}$ and $\mathbf{B}$ are PSD matrices:

For matrix $\mathbf{A}$, for any $\mathbf{x} \in \mathbb{R}^{r_t}$, with the Cauchy–Schwarz inequality, we can obtain

$$\mathbf{x}^\top \mathbf{A}\mathbf{x} = \mathbf{x}^\top(\mathbf{I} - \mathbf{s}\mathbf{s}^\top)\mathbf{x} = \|\mathbf{x}\|_2^2 - (\mathbf{s}^\top\mathbf{x})^2 \geq \|\mathbf{x}\|_2^2 - \|\mathbf{s}\|_2^2\|\mathbf{x}\|_2^2 = 0, \tag{23}$$

note that $\|\mathbf{s}\|_2 = 1$ comes from Lemma 1.

For matrix $\mathbf{B}$, for any $\mathbf{x} \in \mathbb{R}^{r_t}$, we can obtain

$$\mathbf{x}^\top \mathbf{B}\mathbf{x} = \mathbf{x}^\top \mathbf{m}\mathbf{m}^\top\mathbf{x} = \mathbf{m}^\top\mathbf{x} \cdot \mathbf{x}^\top\mathbf{m} = (\mathbf{x}^\top\mathbf{m})^2 \geq 0. \tag{24}$$

Now, we relate the term $\sum_{i,j}(1 - s_i s_j)m_{ii}m_{jj}$ to the trace of the product $\mathbf{A}\mathbf{B}$:

$$\sum_{i,j}(1 - s_i s_j)m_{ii}m_{jj} = \mathbf{m}^\top(\mathbf{I} - \mathbf{s}\mathbf{s}^\top)\mathbf{m} = \mathbf{m}^\top\mathbf{A}\mathbf{m}$$

$$= \mathrm{tr}\left(\mathbf{m}^\top\mathbf{A}\mathbf{m}\right) = \mathrm{tr}\left(\mathbf{A}\mathbf{m}\mathbf{m}^\top\right) = \mathrm{tr}(\mathbf{A}\mathbf{B}). \tag{25}$$

Combine Lemma 2 and Eq. (22), we can obtain

$$\mathbb{E}\left[(\mathrm{tr}(\mathbf{M}_t))^2\right] - \mathbb{E}\left[(\mathrm{tr}(\mathbf{S}_t\mathbf{M}_t)^2\right] = \mathbb{E}\left[\mathrm{tr}(\mathbf{A}\mathbf{B})\right] \geq 0. \tag{26}$$

We focus on the late stages of training near convergence, where the gradient norm becomes very small, often substantially below one (Zhang et al., 2017). Thus, combining Eq. (21), Eq. (26) and Eq. (20), we can obtain:

$$\mathbb{E}\left[(\mathrm{proj}_{\mathrm{Muon}})^2\right] = \mathbb{E}\left[(\mathrm{tr}(\mathbf{M}))^2\right] \geq \mathbb{E}\left[\left(\mathrm{tr}(\mathbf{G}_t^\top\mathbf{V})\right)^2\right] \geq \mathbb{E}\left[(\mathrm{proj}_{\mathrm{SGD}})^2\right]. \tag{27}$$

## C  EXPERIMENTAL SUPPLEMENT

### C.1  ADDITIONAL EXPERIMENTS ON CIFAR

Table 5: Top-1 accuracy (%) (↑) results of different optimizers under various loss functions on CIFAR-10 LT with an imbalance factor of 10. Results for the *Medium* class group are presented as Med. in the table.

| Loss | Method | Many | Med. | Few | All |
|------|--------|------|------|-----|-----|
| CE | SGD | 95.0 | 85.9 | 88.2 | 89.3 |
|  | SAM | 95.2 | 86.3 | 88.1 | 89.1 |
|  | **ProMO** | 95.0 | 86.4 | 89.9 | **90.0** |
|  | **Muon** | 96.1 | 86.3 | 88.3 | 89.8 |
| CB | SGD | 94.9 | 86.4 | 88.4 | 89.6 |
|  | SAM | 95.0 | 86.0 | 87.9 | 89.3 |
|  | **ProMO** | 95.7 | 86.6 | 88.6 | 89.9 |
|  | **Muon** | 95.6 | 87.2 | 88.7 | **90.2** |
| LA | SGD | 93.8 | 87.5 | 92.1 | 90.8 |
|  | SAM | 94.1 | 87.0 | 92.1 | 90.7 |
|  | **ProMO** | 94.5 | 87.5 | 92.2 | 91.0 |
|  | **Muon** | 94.5 | 88.0 | 92.6 | **91.3** |
| BCL | SGD | 94.3 | 87.5 | 91.8 | 90.8 |
|  | SAM | 94.5 | 88.3 | 93.2 | 91.6 |
|  | **ProMO** | 95.0 | 88.6 | 92.5 | **91.7** |
|  | **Muon** | 94.8 | 88.2 | 92.2 | 91.4 |
| ProCo | SGD | 94.8 | 88.6 | 92.6 | 91.7 |
|  | SAM | 94.6 | 88.9 | 93.1 | 91.8 |
|  | **ProMO** | 95.3 | 88.7 | 92.7 | 91.9 |
|  | **Muon** | 94.8 | 88.6 | 93.5 | **92.0** |

Table 5 presents the comparative performance of Muon and ProMO on the CIFAR-10 LT dataset with an imbalance factor of 10. The results demonstrate that both Muon and ProMO consistently surpass the SGD and SAM baselines across various loss functions, aligning with the trend observed in Table 1.

## C.2 Additional Experiments on Computational Overhead

Table 6: Computational overhead of different optimizers under CE loss on long-tailed benchmarks. We report the average training time per epoch (seconds) (↓) and the runtime ratio relative to SGD (in parentheses). Performance of Muon and ProMO are highlighted in blue to group them for focused comparison against the baselines.

| Method | CIFAR-100 | | | | ImageNet-LT | | Places-LT | |
|--------|-----------|---|-----------|---|-------------|---|-----------|---|
| | IF=10 | | IF=100 | | | | | |
| SGD | 3.47s | (1.00×) | 3.10s | (1.00×) | 280.046s | (1.00×) | 224.70s | (1.00×) |
| SAM | 6.94s | (1.99×) | 4.66s | (1.50×) | 392.170s | (1.40×) | 356.08s | (1.58×) |
| **Muon** | 8.22s | (2.37×) | 5.87s | (1.89×) | 435.29s | (1.55×) | 471.63s | (2.10×) |
| **ProMO** | 4.49s | (1.29×) | 3.35s | (1.08×) | 292.51s | (1.04×) | 267.29s | (1.19×) |

In Table 6, we provide additional experiments analyzing computational efficiency. We measure the average training time per epoch across four datasets using the CE loss function. The results show that the SAM optimizer incurs an average of 98% additional training time compared to SGD, while the Muon optimizer increases training time by an average of 106% under the same settings. In contrast, our proposed ProMO increases training time by only 15% on average relative to SGD. These findings are consistent with the results presented in Table 4 and Figs. 3(a) and 3(b).

## C.3 Efficiency Analysis via Gradient Approximation

To further mitigate computational overhead, we investigated reducing the precision of Newton-Schulz orthogonalization as a potential optimization for efficient gradient approximation. Specifically, we evaluated the performance of ProMO on CIFAR-100 LT with CB loss under an imbalance factor of 100, while varying the number of Newton-Schulz iteration steps $N$ from the default 5 down to 2.

The results, summarized in Table 7, demonstrate a clear trade-off between computational cost and accuracy. Consistent with our theoretical complexity analysis (Eq. (6,7)), reducing the iterations successfully lowers the computational overhead. While the default $N = 5$ retains the highest accuracy, we observe that although the total accuracy decreases slightly as $N$ declines, it consistently remains superior to SGD. This validates that gradient approximation via moderately reduced iterations is still an effective method for maintaining robust performance in resource-constrained scenarios.

Table 7: Ablation study on the number of Newton-Schulz iteration steps ($N$) on CIFAR-100 LT under an imbalance factor of 100. We report the top-1 accuracy(%) (↑), the average training time per epoch (seconds) (↓) and the runtime ratio relative to SGD (in parentheses).

| Method | $N$ | Many | Medium | Few | All | Time/epoch |
|--------|-----|------|--------|-----|-----|------------|
| SGD | - | 75.0 | 50.6 | 17.3 | 44.6 | 2.57 (1×) |
| **Muon** | 5 | 76.4 | 53.1 | 19.7 | **46.7** | 4.11 (1.59×) |
| **ProMO** | 5 | 76.5 | 52.5 | 19.3 | 46.4 | 2.94 (1.14×) |
| ProMO | 2 | 76.1 | 51.1 | 17.7 | 45.2 | 2.59 (1.01×) |
| ProMO | 3 | 76.1 | 52.8 | 18.1 | 45.9 | 2.78 (1.09×) |
| ProMO | 4 | 77.2 | 51.3 | 18.8 | 46.1 | 2.82 (1.10×) |

## C.4 Generalization to Natural Language Processing

While our primary evaluation followed mainstream long-tailed learning protocols centered on visual benchmarks (Menon et al., 2021), we further investigated the versatility of Muon by extending our experiments to the Natural Language Processing domain. We conducted experiments using the Yahoo Answers Topic Classification dataset (Zhang et al., 2015). To simulate long-tailed distributions, we constructed two variants by sampling from a 12k training subset with imbalance factors of 10 and 50,

respectively. We divide the classes into *Many*, *Medium*, and *Few* splits, corresponding to the top three, middle four, and bottom three classes sorted by frequency, respectively. Evaluation was performed on a balanced test set containing 4k samples. The model architecture consisted of a fixed pre-trained BERT-base-uncased backbone, followed by an MLP layer and a linear classification head. Both SGD and Muon were trained using CE loss for 20 epochs. As shown in Table 8, Muon consistently outperforms the SGD baseline across different imbalance factors, especially in the tail classes and highly imbalanced setting. This confirms that the benefits of Muon's curvature-aware optimization are not limited to vision tasks but also extend effectively to other modalities like NLP.

Table 8: Top-1 accuracy (%) (↑) results for *Many*, *Medium*, *Few*, and overall classes on long-tailed Yahoo Answers dataset, categorized by imbalance factors (IF) of 10 and 50.

| IF | Method | Many | Medium | Few | All |
|----|--------|------|--------|-----|-----|
| 10 | SGD | 75.9 | 58.5 | 43.3 | 59.1 |
| | **Muon** | 73.0 | 60.9 | 45.6 | **59.9** |
| 50 | SGD | 74.8 | 61.2 | 3.4 | 47.9 |
| | **Muon** | 76.8 | 55.9 | 17.3 | **50.6** |

## C.5 COMPARISON WITH FINE-TUNING METHOD

Recent long-tailed recognition methods have explored fine-tuning paradigms on top of large-scale foundation models, such as LIFT (Shi et al., 2024) and LPT (Dong et al., 2023). To verify that Muon remains effective in this setting, we follow the experimental protocol of LIFT. Specifically, we adopt a pre-trained CLIP ViT-B/16 backbone and fine-tune it on CIFAR-100 LT with IF=100. We adhere to the experimental settings of LIFT for a fair comparison. As shown in Table 9, Muon achieves higher overall accuracy than LIFT, with particularly notable gains on tail classes. This indicates that Muon is complementary to fine-tuning based long-tailed methods, and can further improve representation quality even when starting from strong pre-trained features.

Table 9: Top-1 accuracy (%) (↑) results for *Many*, *Medium*, *Few*, and overall classes on CIFAR-100 LT under IF=100 with a CLIP ViT-B/16 backbone.

| Method | Many | Medium | Few | All |
|--------|------|--------|-----|-----|
| LIFT | 84.4 | 81.1 | 74.4 | 80.2 |
| **Muon** | 85.1 | 81.5 | 76.8 | **81.3** |

## C.6 MUON WITH DECOUPLED TRAINING METHOD

We further explore the performance of Muon when combined with decoupled training methods. Specifically, we evaluate Muon and ProMO under the standard two-stage decoupling framework (Kang et al., 2020): (1) *Stage 1*, trains the backbone representation, and (2) *Stage 2*, re-trains a balanced classifier (cRT) on top of the frozen backbone. Concretely, we train the backbone in Stage 1 using SGD, Muon, or ProMO, and then apply classifier re-training (cRT) in Stage 2. Experiments are conducted on CIFAR-100 LT under IF=10 and IF=100.

As shown in Table 10, Muon and ProMO consistently outperform SGD after cRT, indicating that re-balancing the classifier does not diminish their advantages. Instead, the gains persist because Muon and ProMO improve the quality of learned representations during Stage 1, providing a stronger feature space for the balanced classifier in the later stage.

## C.7 COMPARISON WITH STRONG SAM VARIANT

**Comparison with ImbSAM.** Several SAM variants have been proposed for long-tailed learning, such as ImbSAM (Zhou et al., 2023a). We further conduct additional comparisons to evaluate the effectiveness and efficiency of our proposed ProMO. We compare ProMO against SGD, SAM, and

Table 10: Top-1 accuracy (%) (↑) results for *Many*, *Medium*, *Few*, and overall classes on CIFAR-100 LT, categorized by imbalance factors (IF) of 10 and 100.

| IF | Method | Many | Medium | Few | All |
|----|--------|------|--------|-----|-----|
| 10 | SGD | 67.3 | 61.5 | 55.9 | 61.8 |
| | **Muon** | 68.3 | 61.9 | 57.3 | 62.8 |
| | **ProMO** | 68.9 | 62.4 | 56.5 | **62.9** |
| 100 | SGD | 66.3 | 51.7 | 31.7 | 47.5 |
| | **Muon** | 66.9 | 56.3 | 34.0 | **50.6** |
| | **ProMO** | 66.1 | 54.8 | 34.3 | 50.0 |

ImbSAM on CIFAR-100-LT under imbalance factors IF=10 and IF=100. Following prior work, we consider both CE loss and BCL loss. We also report the training time measured on a single NVIDIA RTX 3090, normalized by the SGD baseline to highlight the efficiency trade-off.

As shown in Table 11, ProMO consistently attains the best overall accuracy across all settings, while being substantially more efficient than SAM and ImbSAM. In particular, ImbSAM requires roughly $2.0\times$-$3.1\times$ the training cost of SGD due to its extra gradient computations, whereas ProMO only incurs a marginal overhead of about $1.1\times$–$1.3\times$. These results demonstrate that ProMO achieves a more favorable accuracy-efficiency trade-off than computationally heavy SAM variants in long-tailed recognition.

Table 11: Top-1 accuracy (%) (↑) results for *Many*, *Medium*, *Few*, and overall classes on CIFAR-100 LT with CE and BCL losses, under imbalance factors (IF) of 10 and 100. We also report the training time (seconds) (↓) and the runtime ratio relative to SGD (in parentheses).

| Loss | IF | Method | Many | Medium | Few | All | Time |
|------|----|--------|------|--------|-----|-----|------|
| CE | 10 | SGD | 75.6 | 62.8 | 48.2 | 60.8 | 696 (1.00×) |
| | | SAM | 76.4 | 64.5 | 49.1 | 61.9 | 1390 (2.00×) |
| | | ImbSAM | 74.0 | 61.4 | 54.6 | 62.4 | 2148 (3.09×) |
| | | **ProMO** | 77.1 | 65.4 | 49.7 | **62.6** | **898** (1.29×) |
| | 100 | SGD | 75.9 | 52.0 | 15.7 | 44.6 | 516 (1.00×) |
| | | SAM | 76.3 | 51.6 | 17.0 | 45.2 | 932 (1.81×) |
| | | ImbSAM | 76.1 | 49.1 | 20.0 | 45.6 | 1270 (2.47×) |
| | | **ProMO** | 77.2 | 53.9 | 16.2 | **45.8** | **670** (1.30×) |
| BCL | 10 | SGD | 71.7 | 64.5 | 59.5 | 64.7 | 1674 (1.00×) |
| | | SAM | 72.5 | 65.2 | 60.0 | 65.3 | 2896 (1.73×) |
| | | ImbSAM | 71.9 | 66.0 | 60.3 | 65.5 | 3482 (2.08×) |
| | | **ProMO** | 73.9 | 66.0 | 60.4 | **66.1** | **1804** (1.08×) |
| | 100 | SGD | 68.5 | 54.2 | 34.2 | 50.5 | 1098 (1.00×) |
| | | SAM | 68.1 | 53.5 | 37.1 | 51.3 | 1802 (1.64×) |
| | | ImbSAM | 68.0 | 52.9 | 40.0 | 52.2 | 2148 (1.96×) |
| | | **ProMO** | 71.1 | 57.5 | 36.3 | **53.1** | **1178** (1.07×) |

**Comparison with LookSAM.** We further compare Muon and ProMO with LookSAM (Liu et al., 2022), a representative efficient SAM variant in balanced scenarios. We compare these methods on CIFAR-100-LT under an imbalance factor of 100. The results are shown in Table 12. Although we verified that LookSAM matches SAM's performance on the balanced CIFAR-100 (both achieve an accuracy of 72.8%), the results show that its accuracy deteriorates substantially on the imbalanced CIFAR-100 LT, particularly on tail classes. In contrast, ProMO retains the robust generalization. This indicates that efficiency techniques effective in balanced settings, such as the gradient decomposition and estimation strategies used in LookSAM, are not robust under severe class imbalance. These findings underscore the value of developing efficient alternatives that remain effective in imbalanced scenarios, such as Muon and ProMO.

Table 12: Top-1 accuracy (%) (↑) results for *Many*, *Medium*, *Few*, and overall classes on CIFAR-100 LT with CE and LA losses, under an imbalance factor of 100. LookSAM-$k$ denotes the method where the SAM update is performed every $k$ steps.

| Loss | Method | Many | Medium | Few | All |
|------|--------|------|--------|-----|-----|
| CE | SAM | 76.3 | 51.6 | 17.0 | 45.2 |
| | **Muon** | 77.2 | 52.4 | 17.3 | 45.8 |
| | **ProMO** | 77.2 | 53.9 | 16.2 | **45.8** |
| | LookSAM-2 | 74.8 | 46.7 | 10.5 | 40.7 |
| | LookSAM-3 | 70.1 | 37.8 | 6.5 | 35.0 |
| | LookSAM-4 | 64.2 | 29.9 | 4.6 | 30.1 |
| LA | SAM | 75.4 | 50.6 | 19.0 | 45.4 |
| | **Muon** | 76.4 | 52.2 | 19.7 | **46.5** |
| | **ProMO** | 76.5 | 52.5 | 19.3 | 46.4 |
| | LookSAM-2 | 73.2 | 46.9 | 13.6 | 41.5 |
| | LookSAM-3 | 70.0 | 40.1 | 9.7 | 36.9 |
| | LookSAM-4 | 67.3 | 35.8 | 7.8 | 34.1 |

## C.8 RESULTS ON LARGE-SCALE REAL-WORLD LONG-TAILED DATASET

To provide a more comprehensive evaluation, we extend our experiments to the large-scale real-world setting. We additionally benchmark our method on the iNaturalist-2018 (Horn et al., 2018) dataset. The iNaturalist-2018 is a large-scale real-world long-tailed dataset that contains 437.5k training images from 8,142 species. Following mainstream protocols (Cui et al., 2019; Du et al., 2024), we adopt a ResNet-50 backbone trained for 90 epochs using CE loss. We compare SGD, SAM, Muon, and our proposed ProMO under the *Many*, *Medium*, and *Few* splits. As shown in Table 13, Muon and ProMO both outperform SGD and SAM across all class splits, with especially clear improvements on tail classes. These results demonstrate that curvature-aware optimization methods, such as Muon and ProMO, generalize effectively to more challenging large-scale long-tailed datasets.

Table 13: Top-1 accuracy (%) (↑) results for *Many*, *Medium*, *Few*, and overall classes on iNaturalist-2018 dataset. Muon and ProMO exhibit consistent improvements across all class splits.

| Method | Many | Medium | Few | All |
|--------|------|--------|-----|-----|
| SGD | 74.6 | 64.9 | 56.8 | 62.7 |
| SAM | 76.4 | 66.8 | 58.8 | 64.6 |
| **ProMO** | 77.5 | 67.9 | 59.8 | 65.7 |
| **Muon** | 78.0 | 68.4 | 60.6 | **66.3** |

## C.9 DEEPER UNDERSTANDING BETWEEN THEORETICAL ANALYSIS AND PROMO DESIGN

To demonstrate that the design of ProMO is grounded in the intrinsic training dynamics of long-tailed learning, we conducted an empirical analysis tracking the evolution of loss landscape geometry. Specifically, we monitored the Hessian trace of the least frequent class on CIFAR-100 LT under IF=100 across the training process for SGD, Muon, and ProMO. The results are presented in Table 14.

These results show that, in the early training phase, both SGD and Muon exhibit relatively low traces. This suggests that during early exploration, the inherent stochasticity of SGD gradients provides sufficient noise to avoid sharp minima. Consequently, the complex orthogonalization operations of Muon incur computational overhead without offering significant geometric advantages during this period. This justifies ProMO's design choice to prioritize SGD in the early stages to maintain high computational efficiency while the optimization landscape is still being actively explored.

In later training stages, especially as the model nears convergence, the trace for SGD increases dramatically, indicating convergence to a sharp minimum, which is known to harm tail-class generalization.

Table 14: Evolution of the Hessian trace on CIFAR-100 LT under imbalance factor of 100. Lower values indicate flatter minima, which correlate with better generalization.

| Epoch | SGD | Muon | ProMO |
|---|---|---|---|
| 80 | 664.6 | 636.1 | 691.5 |
| 120 | 956.1 | 450.8 | 615.2 |
| 180 | 1629.6 | 703.6 | 667.6 |
| 200 | 2137.5 | 536.0 | 514.1 |

In contrast, Muon maintains significantly lower trace values, validating our theoretical analysis that it effectively escapes sharp regions by amplifying updates along negative curvature directions. This provides the motivation for ProMO to progressively increase its usage of the Muon optimizer as training advances. These observations confirm ProMO as a principled solution that combines SGD's early efficiency with Muon's capability to escape sharp minima in the later stages.

## D  DISCUSSIONS

**Additional related work.** Recent advancements in long-tailed recognition have diversified beyond traditional re-balancing techniques. In the realm of contrastive learning, GPaCo (Cui et al., 2024) identifies the bias of supervised contrastive loss towards high-frequency classes and introduces parametric learnable centers to rebalance optimization dynamics. For handling diverse test distributions, DirMixE (Yang et al., 2024) proposes a sophisticated mixture-of-experts strategy based on Dirichlet meta-distributions to capture both global and local label distribution variations. Furthermore, addressing the geometry of the loss landscape has become a pivotal direction; CC-SAM (Zhou et al., 2023b) argues that naive flattening is insufficient for long-tailed learning and proposes a class-conditional sharpness-aware minimization to robustify the classifier against parameter perturbations.

**Muon enhances representation learning.** Muon fundamentally improves representation learning by guiding optimization toward flatter minima. In long-tailed recognition, a critical representational failure mode is the tendency for minority classes to converge to sharp regions of the loss landscape, which undermines generalization capabilities. Muon addresses this challenge. By amplifying updates along directions of negative curvature via gradient orthogonalization, Muon facilitates the escape from these sharp regions and guides optimization toward flatter solutions. Securing these flatter minima, which is evidenced by improved loss-landscape metrics on tail classes, is essential for learning robust representations that generalize better to underrepresented data, going beyond mere improvements in convergence speed. Table 9 and Table 10 also provide more empirical evidence that Muon could improve the quality of learned representations.

**Regarding CNC assumption.** The Correlated Negative Curvature (CNC) assumption is well established in the non-convex optimization literature. It has been theoretically justified for learning half-spaces Daneshmand et al. (2018), while subsequent studies, such as (Wang et al., 2020)), have provided further validation through extensive analyses on deep networks of varying widths and depths. It has also been adopted in broader domains such as manifold optimization (Criscitiello & Boumal, 2019), making it a standard and widely accepted assumption. Importantly for our setting, CNC has been examined directly in long-tailed learning. Rangwani et al. (2022) shows that tail classes exhibit strong negative curvature that traps SGD in sharp minima, and that methods like SAM alleviate this issue. Later works such as Zhou et al. (2023a) further support these observations. Our analysis builds on this established foundation and does not require stronger assumptions than prior work.

**Targeted design addressing long-tailed learning challenges.** Our work identifies and operationalizes a unique complementarity between gradient orthogonalization and long-tailed learning through two specialized contributions. First, our theoretical analysis demonstrates that Muon specifically addresses optimization bottlenecks inherent to underrepresented data by enhancing updates along directions of negative curvature, enabling the model to escape sharp minima that hinder tail-class generalization. Second, building on this insight, we designed ProMO specifically for long-tailed training dynamics. ProMO progressively integrates Muon during critical later stages when SGD fails to escape sharp minima, thereby increasing tail-class generalization while mitigating computational overhead. This provides an efficient solution essential for large-scale imbalanced benchmarks.

**ProMO as a trade-off between efficiency and generalization.** Our primary design objective of ProMO is to approximate Muon's generalization benefits while substantially mitigating its computational overhead. As shown in Table 1 and Table 2, ProMO exhibits a consistent performance pattern, reliably outperforming SGD and remaining competitive with Muon in accuracy. Crucially, as shown in Table 4, it achieves these results while drastically reducing training costs compared to the significant overhead demands of Muon and SAM. Thus, ProMO successfully delivers its intended precise trade-off between high efficiency and robust tail-class generalization.

## LLM USAGE

We thank GPT[1] and Gemini[2] for their assistance in language polishing when writing the paper. The authors take full responsibility for all the content of this paper.

---

[1] https://chatgpt.com
[2] https://gemini.google.com

