# OpenReview forum: "Long-tailed Learning with Muon Optimizer"
_ICLR.cc/2026/Conference — Submitted to ICLR 2026_

### Official Review · Reviewer_MTSZ · 2025-10-27

**Soundness:** 3
**Presentation:** 3
**Contribution:** 2
**Rating:** 4
**Confidence:** 4

**Summary:**

This paper investigates long-tailed learning from the perspective of the sharpness of the loss landscape. It argues that previous sharpness-aware minimization (SAM) methods and their variants incur high computational costs. Similarly, although the Muon optimizer can help models escape sharp regions, it is also computationally expensive. To address these issues, the paper proposes ProMO, a method that applies the Muon optimizer following a sinusoidal scheduling strategy to balance effectiveness and efficiency. Experimental results on several mainstream long-tailed datasets demonstrate the effectiveness of the proposed approach.

**Strengths:**

1. The paper is well-organized, and the overall writing flow is clear and logical, making it easy to follow.
2. The proposed solution is simple yet technically sound.
3. The theoretical and computational analyses provide valuable insights that strengthen the motivation and understanding of the proposed approach.

**Weaknesses:**

1. The technical novelty of the proposed method appears limited, as it primarily combines Muon and SGD in a hybrid scheduling manner. To strengthen the contribution, the authors are encouraged to provide deeper insights or analyses on the underlying training dynamics that motivate this design.
2. The paper does not include comparisons with several strong SAM variants, such as ImbSAM [1] and CC-SAM [2], which are highly relevant baselines in this context.
3. Experimental results on the iNaturalist-2018 dataset are missing, which limits the completeness of the evaluation.
4. According to the reported results, ProMO exhibits inconsistent performance improvements across different datasets.

Reference:

[1] ImbSAM: ACloser Look at Sharpness-Aware Minimization in Class-Imbalanced Recognition. ICCV 2023.

[2] Class-Conditional Sharpness Aware Minimization for Deep Long-Tailed Recognition. CVPR 2023.

**Questions:**

Please refer to the Weaknesses.

---

> ### Author Response · Authors · 2025-11-24
> **Response to Reviewer MTSZ (1/3)**
>
> > The technical novelty of the proposed method appears limited, as it primarily combines Muon and SGD in a hybrid scheduling manner. To strengthen the contribution, the authors are encouraged to provide deeper insights or analyses on the underlying training dynamics that motivate this design.
>
> Thank you for this constructive comment. We would like to clarify that our contribution extends beyond a simple combination of optimizers. Our method is grounded in a specific analysis of how optimization geometry interacts with class imbalance, leading to the following novel insights regarding training dynamics:
>
> **Gradient orthogonalization as a mechanism to improve tail-class generalization**: Our technical novelty stems from identifying a theoretical alignment between Muon’s update rule and the specific topology of long-tailed loss landscapes. We provide theoretical analysis in Theorem 1, demonstrating that its gradient orthogonalization increases the update component along directions of negative curvature, making it more effective than SGD at escaping the sharp minima that predominantly affect tail classes. This is corroborated by our class-wise Hessian study and eigenvalue spectra on long-tailed benchmarks, which show that tail classes are indeed associated with sharper regions and that Muon converges to significantly flatter minima, leading to improved tail generalization.
>
> **Dynamics-aware probability scheduling for accuracy-efficiency trade-off**: We introduce ProMO as a dynamic adaptation rather than a static mixture, motivated by prior work showing that SGD’s early stochasticity helps escape sharp regions while its post-decay behavior becomes increasingly deterministic and prone to stalling. Our sinusoidal design is engineered to minimize computational overhead by utilizing SGD during the exploration phase and progressively shifting to Muon only as the model approaches convergence, precisely when the exploitation of negative curvature is most beneficial for tail generalization. Our ablations against various alternating schedules confirm that this dynamics-aware approach yields the optimal trade-off between training efficiency and tail-class accuracy.
>
> We appreciate your helpful suggestions and have included these discussions in the Discussion section of the revised version.

---

> ### Author Response · Authors · 2025-11-24
> **Response to Reviewer MTSZ (2/3)**
>
> > The paper does not include comparisons with several strong SAM variants, such as ImbSAM [1] and CC-SAM [2], which are highly relevant baselines in this context.
>
> Thank you for this valuable suggestion. To address your concern, we have conducted additional experiments to compare our proposed ProMO with ImbSAM [a] on the CIFAR-100-LT dataset. We evaluated the performance under imbalance factors of IF=10 and IF=100, using both CE and BCL losses. Furthermore, we recorded the training time (s) on an NVIDIA RTX 3090 to highlight the efficiency trade-offs. The results are summarized in the tables below:
>
> | Method | Many  | Medium  | Few   | All   | Time    |
> |--------|:-----:|:-----:|:-----:|:-----:|:-------------:|
> | SGD    | 75.6  | 62.8  | 48.2  | 60.8  | 696(1×)     |
> | SAM    | 76.4  | 64.5  | 49.1  | 61.9  | 1390(2.00×) |
> | ImbSAM | 74.0  | 61.4  | 54.6  | 62.4  | 2148(3.09×) |
> | ProMO  | 77.1  | 65.4  | 49.7  | 62.6  | **898(1.29×)**  |
>
> *Tab.(a) Comparison with ImbSAM with CE under IF=10.*
>
> | Method | Many  | Medium | Few   | All   | Time    |
> |--------|:-----:|:-----:|:-----:|:-----:|:-----------:|
> | SGD    | 75.9  | 52.0  | 15.7  | 44.6  | 516(1×)     |
> | SAM    | 76.3  | 51.6  | 17.0  | 45.2  | 932(1.81×)  |
> | ImbSAM | 76.1  | 49.1  | 20.0  | 45.6  | 1270(2.47×) |
> | ProMO  | 77.2  | 53.9  | 16.2  | 45.8  | **670(1.30×)**  |
>
> *Tab.(b) Comparison with ImbSAM with CE under IF=100.*
>
> | Method | Many  | Medium  | Few   | All   | Time    |
> |--------|:-----:|:-----:|:-----:|:-----:|:-----------:|
> | SGD    | 71.7  | 64.5  | 59.5  | 64.7  | 1674(1×)    |
> | SAM    | 72.5  | 65.2  | 60.0  | 65.3  | 2896(1.73×) |
> | ImbSAM | 71.9  | 66.0  | 60.3  | 65.5  | 3482(2.08×) |
> | ProMO  | 73.9  | 66.0  | 60.4  | 66.1  | **1804(1.08×)** |
>
> *Tab.\(c\) Comparison with ImbSAM with BCL under IF=10.*
>
> | Method | Many  | Medium  | Few   | All   | Time    |
> |--------|:-----:|:-----:|:-----:|:-----:|:-----------:|
> | SGD    | 68.5  | 54.2  | 34.2  | 50.5  | 1098(1×)    |
> | SAM    | 68.1  | 53.5  | 37.1  | 51.3  | 1802(1.64×) |
> | ImbSAM | 68.0  | 52.9  | 40.0  | 52.2  | 2148(1.96×) |
> | ProMO  | 71.1  | 57.5  | 36.3  | 53.1  | **1178(1.07×)** |
>
> *Tab.(d) Comparison with ImbSAM with BCL under IF=100.*
>
> The experimental results demonstrate the advantages of ProMO over ImbSAM in terms of both **overall performance** and **computational efficiency**. Specifically, ProMO consistently outperforms ImbSAM in total accuracy across various settings. Regarding computational efficiency, a critical limitation of ImbSAM is its high cost due to the requirement for more gradient calculations than standard SAM. As observed in the tables, ImbSAM requires approximately 2.0× to 3.1× the training time of SGD. In contrast, our ProMO is significantly more efficient, with only a marginal increase in cost (approximately 1.1× to 1.3× that of SGD), making it a much more practical solution for long-tailed learning than computationally heavy SAM variants.
>
> Thank you again for this insightful comment. We have included the details of these comparative experiments in the Experimental Supplement section to provide a more comprehensive evaluation.
>
> [a] ImbSAM: A Closer Look at Sharpness-Aware Minimization in Class-Imbalanced Recognition, ICCV 2023.
>
> [b] Class-Conditional Sharpness-Aware Minimization for Deep Long-Tailed Recognition, CVPR 2023.
>
> > Experimental results on the iNaturalist-2018 dataset are missing, which limits the completeness of the evaluation.
>
> Thank you for this constructive suggestion. To address your concern, we conducted additional experiments on the iNaturalist-2018 dataset. Following the mainstream evaluation protocols [a, b], we employed a ResNet-50 backbone and trained the model for 90 epochs with CE loss. The comparison results between SGD, SAM, Muon and ProMO are presented in the table below:
>
> | Method | Many | Medium | Few  |  All  |
> | ------ |:----:|:------:|:----:|:-----:|
> | SGD    | 74.6 | 64.9   | 56.8 | 62.7  |
> | SAM    | 76.4 | 66.8   | 58.8 | 64.6  |
> | ProMO  | 77.5 | 67.9   | 59.8 | 65.7  |
> | Muon   | 78.0 | 68.4   | 60.6 | 66.3  |
>
> The table shows that Muon and ProMO outperform both the SGD and SAM optimizers. Notably, they exhibit significant superiority in the tail classes. These findings on a large-scale real-world dataset align well with the theoretical analysis and the trends observed in Table 1 and 2. We have included the experiment details in the revised version to ensure a more thorough evaluation.
>
> [a] Class-Balanced Loss Based on Effective Number of Samples, CVPR 2019.
>
> [b] Probabilistic Contrastive Learning for Long-Tailed Visual Recognition, TPAMI 2024.

---

> ### Author Response · Authors · 2025-11-24
> **Response to Reviewer MTSZ (3/3)**
>
> > According to the reported results, ProMO exhibits inconsistent performance improvements across different datasets.
>
> Thank you for this observation. We would like to clarify that our primary goal with ProMO is not to uniformly dominate the Muon optimizer across every dataset and loss, but rather to approximate Muon’s generalization benefits while substantially mitigating its computational overhead. At this stage, ProMO exhibits a highly consistent performance pattern: it reliably outperforms SGD and often remains close to Muon in terms of accuracy, as shown in Table 1 and 2. Crucially, ProMO achieves these competitive results while drastically reducing training costs. We summarize representative training costs relative to SGD (e.g., with ProCo) in Table 4 below:
>
> | Method    | CIFAR-100 LT IF=10 | CIFAR-100 LT IF=100 | ImageNet-LT     | Places-LT       |
> | --------- | :----------------: | :-----------------: | :-------------: | :-------------: |
> | SGD       | 1.00×      | 1.00×        | 1.00×  | 1.00×  |
> | SAM       | 1.92×      | 1.85×        | 2.73×  | 4.54×  |
> | Muon      | 1.59×      | 1.60×        | 1.63×  | 2.58×  |
> | ProMO     | **1.12×**  | **1.17×**   | **1.17×** | **1.40×** |
>
> As shown in the table, ProMO requires only **~1.1–1.4×** the training time of SGD, whereas Muon demands **~1.6–2.6×** and SAM requires **~1.9–4.5×**. This evidence confirms that ProMO delivers the precise trade-off between high efficiency and robust tail-class generalization that it was designed for. In the revised version, we have clarified this design objective and added a brief discussion in the Discussion section to make this point more explicit.

---

> > ### Comment · Reviewer_MTSZ · 2025-11-25
> >
> > Thank you for the authors’ efforts and for providing additional results. While some of my concerns have been partially addressed, most key issues remain unresolved:
> >
> > - I acknowledge the paper’s theoretical contributions, though they are unsurprising given existing literature. More importantly, the proposed solution appears largely heuristic. This is precisely why I encourage the authors to provide stronger empirical or theoretical insights explaining why the proposed scheduling mechanism improves optimization, and how it connects to the theoretical motivations behind the Muon optimizer.
> > Currently, the connection between the theoretical analysis and the algorithmic design is weak.
> >
> > - The results on iNaturalist-2018 are not competitive.
> > The authors should clearly state which baseline implementation was used for this dataset. Moreover, across all benchmarks, the proposed method does not reach state-of-the-art performance. If the main intended contribution is to show that Muon/ProMu is an efficient and effective alternative to SAM-based approaches for long-tailed recognition, then direct comparison against SAM variants, particularly those efficient ones (e.g., [1][2]), becomes essential.
> > Without this comparison, it is difficult to conclude the superiority or practical advantage of the proposed approach.
> >
> > I also have carefully read the other reviewer's comments and find we might share some similar concerns. Anyway, I am open to discussion.
> >
> >
> > Reference:
> >
> > [1] An Adaptive Policy to Employ Sharpness-Aware Minimization. ICLR 2023.
> >
> > [2] Towards Efficient and Scalable Sharpness-Aware Minimization. CVPR 2022.

---

> > > ### Author Response · Authors · 2025-12-03
> > > **Response to Reviewer MTSZ (1/2)**
> > >
> > > > I acknowledge the paper’s theoretical contributions, though they are unsurprising given existing literature. More importantly, the proposed solution appears largely heuristic. This is precisely why I encourage the authors to provide stronger empirical or theoretical insights explaining why the proposed scheduling mechanism improves optimization, and how it connects to the theoretical motivations behind the Muon optimizer. Currently, the connection between the theoretical analysis and the algorithmic design is weak.
> > >
> > > Thank you for your valuable and constructive feedback. To further address your concern and validate that ProMO is not merely a heuristic but a method grounded in training dynamics, we conducted an additional experiment to track the evolution of the loss landscape geometry throughout the training process. Specifically, we monitored the Hessian trace of the least frequent class on CIFAR-100 LT under IF=100 for SGD, Muon, and ProMO. This quantifies the sharpness of the loss landscape, where lower values indicate flatter, more generalizable minima. The results are presented below:
> > >
> > > | Epoch | SGD    | Muon  | ProMO |
> > > | ----- | ------ | ----- | ----- |
> > > | 80    | 664.6  | 636.1 | 691.5 |
> > > | 120   | 956.1  | 450.8 | 615.2 |
> > > | 180   | 1629.6 | 703.6 | 667.6 |
> > > | 200   | 2137.5 | 536.0 | 514.1 |
> > >
> > > These results demonstrate two key insights that motivate the design of ProMO:
> > >
> > > **Advantage of Muon During Late Training.** In later training stages, especially as the model nears convergence, the trace for SGD increases dramatically, indicating convergence to a sharp minimum, which could harm the generalization of tail classes. In contrast, Muon maintains significantly lower traces, validating its ability to locate flatter minima. This aligns with our theoretical analysis that Muon escapes sharp regions by amplifying updates along negative curvature directions. This provides the motivation for ProMO to progressively increase its usage of the Muon optimizer as training advances.
> > >
> > > **Sufficiency of SGD During Early Exploration.** In the early training phase, both SGD and Muon exhibit relatively low Hessian traces. This suggests that during the initial, more exploratory phase of training, the inherent gradient stochasticity is sufficient to avoid sharp minima. During this period, Muon’s complex orthogonalization incurs overhead without offering significant advantages over the highly efficient SGD. This justifies ProMO’s design, which prioritizes SGD early to maintain efficiency while the optimization landscape is still being actively explored.
> > >
> > > We believe these results substantiate the connection between our theoretical analysis and algorithmic design. They confirm ProMO as a principled solution that combines SGD’s early efficiency with Muon’s capability to escape sharp minima in the later stages. We thank you again for your insightful feedback and we have included this analysis and discussion in the revised version.

---

> > > ### Author Response · Authors · 2025-12-03
> > > **Response to Reviewer MTSZ (2/2)**
> > >
> > > > The results on iNaturalist-2018 are not competitive. The authors should clearly state which baseline implementation was used for this dataset. Moreover, across all benchmarks, the proposed method does not reach state-of-the-art performance. If the main intended contribution is to show that Muon/ProMO is an efficient and effective alternative to SAM-based approaches for long-tailed recognition, then direct comparison against SAM variants, particularly those efficient ones (e.g., [1][2]), becomes essential. Without this comparison, it is difficult to conclude the superiority or practical advantage of the proposed approach.
> > >
> > > Thank you for your insightful feedback and valuable suggestion. For the implementation on iNaturalist-2018, we followed the training framework in [1] and implemented SAM based on [2]. Indeed, the intended contribution of ProMO is to deliver a precise balance between high efficiency and the robust tail-class generalization offered by Muon. To further address your concerns, we conducted additional experiments comparing Muon and ProMO with an efficient SAM variant, LookSAM [3], with both CE and LA loss, on CIFAR-100-LT under IF = 100. The results are summarized below:
> > >
> > > | Method       | Many  | Medium | Few   | All   |
> > > |--------------|-------|--------|-------|-------|
> > > | SAM          | 76.3  | 51.6   | 17.0  | 45.2  |
> > > | Muon         | 77.2  | 52.4   | 17.3  | 45.8  |
> > > | ProMO     | 77.2  | 53.9   | 16.2  | 45.8  |
> > > | LookSAM-2 | 74.8  | 46.7   | 10.5  | 40.7  |
> > > | LookSAM-3 | 70.1  | 37.8   | 6.5   | 35.0  |
> > > | LookSAM-4 | 64.2  | 29.9   | 4.6   | 30.1  |
> > >
> > > *Tab.(a) Comparison with LookSAM with CE loss.*
> > >
> > > | Method       | Many  | Medium | Few   | All   |
> > > |--------------|-------|--------|-------|-------|
> > > | SAM          | 75.4  | 50.6   | 19.0  | 45.4  |
> > > | Muon         | 76.4  | 52.2   | 19.7  | 46.5  |
> > > | ProMO     | 76.5  | 52.5   | 19.3  | 46.4  |
> > > | LookSAM-2 | 73.2  | 46.9   | 13.6  | 41.5  |
> > > | LookSAM-3 | 70.0  | 40.1   | 9.7   | 36.9  |
> > > | LookSAM-4 | 67.3  | 35.8   | 7.8   | 34.1  |
> > >
> > > *Tab.(b) Comparison with LookSAM with LA loss.*
> > >
> > > Although we verified that LookSAM matches SAM’s performance on the balanced CIFAR-100 in our implementation (both achieve an accuracy of 72.8%), the results show that its accuracy deteriorates substantially on the imbalanced CIFAR-100 LT, particularly on tail classes. Moreover, as the frequency of SAM steps decreases to improve efficiency (from LookSAM-2 to LookSAM-4), the accuracy drops dramatically. This indicates that efficiency techniques effective in balanced settings, such as the gradient decomposition and estimation strategies used in LookSAM, are not robust under severe class imbalance. These findings underscore the value of developing efficient alternatives that remain effective in imbalanced scenarios, such as Muon and ProMO. We have included these experimental details in the Experimental Supplement section of the revised version. Thank you again for this valuable suggestion, which has substantially enriched our analysis.
> > >
> > > [1] Probabilistic Contrastive Learning for Long-Tailed Visual Recognition, TPAMI 2024.
> > >
> > > [2] Escaping Saddle Points for Effective Generalization on Class-Imbalanced Data, NeurIPS 2022.
> > >
> > > [3] Towards Efficient and Scalable Sharpness-Aware Minimization, CVPR 2022.

---

### Official Review · Reviewer_ZKRh · 2025-10-27

**Soundness:** 3
**Presentation:** 3
**Contribution:** 2
**Rating:** 6
**Confidence:** 4

**Summary:**

This paper addresses the critical challenge of long-tailed learning from an optimization perspective by leveraging the Muon optimizer.
The main contributions are as follows:

1. It demonstrates that Muon’s gradient orthogonalization enhances updates along negative curvature directions, enabling the optimizer to escape sharp minima more effectively.

2. It introduces the Progressive Muon Optimizer (ProMO), a hybrid approach that dynamically alternates between SGD and Muon using a sinusoidal probability schedule to balance performance and computational cost.

In summary, the proposed Muon and ProMO optimizers show potential as replacements for conventional SGD-based methods, achieving improved overall performance at the cost of higher computational overhead.

**Strengths:**

1. The paper presents a solid theoretical analysis of the Muon optimizer, with Theorem 1 proving that it amplifies gradient projections along negative curvature directions under the Correlated Negative Curvature (CNC) assumption. This provides a sound theoretical foundation for understanding how Muon effectively escapes sharp minima.

2. The experimental evaluation is comprehensive and convincing, covering a wide range of mainstream loss functions and optimization algorithms.

**Weaknesses:**

1. The Muon optimizer appears to be a general optimization method applicable to existing models, but this paper lacks specific algorithmic design or adaptation specialized for long-tailed learning.

2. The comparison of fine-tuning strategies is missing. For example, recent methods such as LIFT and LPT are not included.
In addition, the paper does not consider decoupled training strategies.
It is also unclear whether using a re-balancing classifier would offset the reported improvements.

3. The Muon optimizer sometimes incurs higher computational overhead than SAM and does not consistently achieve superior performance, which may limit its practical effectiveness. (Nevertheless, the proposed ProMO variant shows better efficiency and performance trade-offs. Why not directly treat ProMO as the main part of the proposed mehtod rather than as an auxiliary extension?)

4. The paper does not provide code or implementation details, which prevents further verification and reproducibility of the proposed method.

**Questions:**

1. What does the deep blue line in the tables represent?

2. Will the authors release the code?

---

> ### Author Response · Authors · 2025-11-24
> **Response to Reviewer ZKRh (1/3)**
>
> > The Muon optimizer appears to be a general optimization method applicable to existing models, but this paper lacks specific algorithmic design or adaptation specialized for long-tailed learning.
>
> Thank you for your insightful comment. We respectfully clarify that our contribution extends beyond the simple application of a general optimizer. Our work identifies a unique complementarity between gradient orthogonalization and long-tailed learning, which we have operationalized through the following two specialized contributions:
>
> **Theoretical alignment with tail-class optimization**: We provide a novel theoretical analysis demonstrating that Muon’s gradient orthogonalization specifically addresses the optimization bottlenecks inherent to long-tailed data. Prior research and our empirical evidence show that tail classes disproportionately converge to sharp minima, leading to poor generalization. We prove that Muon effectively enhances updates along directions of negative curvature, enabling the model to escape these specific sharp regions, which is a capability that is critical for tail classes in long-tailed learning.
>
> **Specialized algorithmic design via ProMO**: Building on the above insight, we designed ProMO specifically for the training dynamics of long-tailed learning. Since SGD provides sufficient noise in early training stages but fails to escape sharp tail-class minima as the learning rate decays, ProMO focuses on progressively integrating Muon during these critical later stages. This specialized design increases tail-class generalization while significantly mitigating the computational overhead of the original Muon, providing a computationally efficient solution essential for large-scale imbalanced benchmarks.
>
> We believe these points demonstrate that our approach is not merely a general optimization study, but a targeted solution for the specific geometric and dynamic challenges of long-tailed learning. We have included this discussion in the Discussion section of the revised version.

---

> ### Author Response · Authors · 2025-11-24
> **Response to Reviewer ZKRh (2/3)**
>
> > The comparison of fine-tuning strategies is missing. For example, recent methods such as LIFT and LPT are not included. In addition, the paper does not consider decoupled training strategies. It is also unclear whether using a re-balancing classifier would offset the reported improvements.
>
> Thank you for the insightful comments. We agree that incorporating these comparisons would strengthen the comprehensiveness of our evaluation. To address your concerns, we have conducted additional experiments on both fine-tuning method and decoupled training. The detailed results and analyses are provided below.
>
> **Comparison with fine-tuning method.** To demonstrate the effectiveness of Muon in fine-tuning scenarios, we follow the experimental setting of LIFT [a]. We used a pre-trained CLIP ViT-B/16 as the backbone and performed fine-tuning on CIFAR-100LT (IF=100). We compared the performance of the SGD optimizer against our Muon optimizer. The results are summarized in the table below:
>
> | Method | Many | Medium | Few  | All |
> | ------ | :--: | :----: | :--: | :---: |
> | LIFT   | 84.4 | 81.1   | 74.4 | 80.2  |
> | Muon   | 85.1 | 81.5   | 76.8 | 81.3  |
>
> *Tab.(a) Comparison with fine-tuning method.*
>
> The results show that Muon outperforms LIFT in overall accuracy. Notably, Muon demonstrates a significant advantage in the tail classes. This result is consistent with the main trends reported in our paper.
>
> **Comparison with decoupled training method.** We adopt the standard decoupling framework [b]. We trained the backbone using SGD, Muon, and ProMO in Stage 1, and subsequently applied classifier Re-Training (cRT) in Stage 2 to re-balance the classifier. We conducted these experiments on CIFAR-100LT under Imbalance Factors (IF) of 10 and 100. The results are summarized in the tables below:
>
> | Method | Many  | Medium  | Few   | All   |
> |--------|:-----:|:-----:|:-----:|:-----:|
> | SGD    | 67.3  | 61.5  | 55.9  | 61.8  |
> | Muon   | 68.3  | 61.9  | 57.3  | 62.8  |
> | ProMO  | 68.9  | 62.4  | 56.5  | 62.9  |
>
> *Tab.(b) Comparison with decoupled method under IF=10.*
>
> | Method | Many  | Medium  | Few   | All   |
> |--------|:-----:|:-----:|:-----:|:-----:|
> | SGD    | 66.3  | 51.7  | 31.7  | 47.5  |
> | Muon   | 66.9  | 56.3  | 34.0  | 50.6  |
> | ProMO  | 66.1  | 54.8  | 34.3  | 50.0  |
>
> *Tab.\(c\) Comparison with decoupled method under IF=100.*
>
>
> The results indicate that using a re-balancing classifier does not offset the improvements gained from Muon and ProMO. Muon and ProMO achieve an overall accuracy higher than SGD. This demonstrates that Muon and ProMO facilitate the learning of a higher-quality feature representation during the backbone training stage (Stage 1). This improved feature space provides a better foundation for the re-balancing classifier in Stage 2.
>
> We have included a discussion on fine-tuning methods (e.g., LIFT [a], LPT [c]) and decoupled training methods in the Related Work section, and have added the details of these experiments in the Experimental Supplement section of the revised version.
>
> [a] Long-Tail Learning with Foundation Model: Heavy Fine-Tuning Hurts, ICML 2024.
>
> [b] Decoupling Representation and Classifier for Long-Tailed Recognition, ICLR 2020.
>
> [c] LPT: Long-tailed Prompt Tuning for Image Classification, ICLR 2023.
>
>
> > The Muon optimizer sometimes incurs higher computational overhead than SAM and does not consistently achieve superior performance, which may limit its practical effectiveness. (Nevertheless, the proposed ProMO variant shows better efficiency and performance trade-offs. Why not directly treat ProMO as the main part of the proposed method rather than as an auxiliary extension?)
>
> Thanks for your assessment and we greatly appreciate this constructive suggestion. Indeed, while the standard Muon optimizer provides theoretical benefits, its computational cost limits its scalability in practical long-tailed scenarios. In fact, our research trajectory began with a deep theoretical investigation into understanding Muon, specifically, identifying its unique capability to escape the sharp minima prevalent in tail classes via gradient orthogonalization. It was this theoretical insight that revealed when Muon is most effective: during the later training stages where SGD often stalls in sharp regions. This understanding directly motivated the design of ProMO, which dynamically integrates Muon only when necessary. We have revised parts of the Introduction and Method sections to present ProMO as the primary proposed method, with the analysis of Muon serving as the theoretical foundation that justifies ProMO’s specific design and effectiveness.

---

> ### Author Response · Authors · 2025-11-24
> **Response to Reviewer ZKRh (3/3)**
>
> > The paper does not provide code or implementation details, which prevents further verification and reproducibility of the proposed method. Will the authors release the code?
>
> Thank you for your valuable feedback. We are fully committed to ensuring that our proposed method is verifiable. To this end, we have included the source code in the supplementary materials. After the anonymous review process, we will release a public repository with more detailed comments, examples, and documentation to further facilitate easy replication of our results.
>
> > What does the deep blue line in the tables represent?
>
> Thank you for your valuable feedback. Both our ProMO and the Muon optimizer are highlighted in blue to visually group the Muon family of optimizers, which is the core subject of our study. This design facilitates a focused comparison against the baselines (SGD and SAM) and makes their shared performance advantage more apparent. Thanks again and we have carefully revised the table captions to clearly state this for future readers.

---

> > ### Comment · Reviewer_ZKRh · 2025-11-25
> >
> > Thanks for the rebuttal. I believe the authors have addressed my concerns, and I would like to maintain my original rating. I will also wait for the other reviewers’ feedback before any further discussion.

---

### Official Review · Reviewer_7gu8 · 2025-10-28

**Soundness:** 3
**Presentation:** 3
**Contribution:** 2
**Rating:** 6
**Confidence:** 4

**Summary:**

The paper tackles the challenge of poor generalization in tail classes within long-tailed learning by examining the optimization process through the lens of loss landscape geometry. It demonstrates that tail classes are often prone to converging to sharper minima in the loss landscape. The authors further show that the recently proposed Muon optimizer enhances gradient projections along directions of negative curvature, enabling faster escape from sharp minima. However, Muon introduces considerable computational overhead. To address this issue, the paper proposes ProMO, a progressive hybrid optimizer that alternates between SGD and Muon according to a sinusoidal probability schedule, effectively balancing computational efficiency and generalization. Extensive experiments validate the effectiveness of the proposed method.

**Strengths:**

- The paper is generally well-written and easy to follow.
- This paper explores the application of the Muon optimizer in long-tailed learning and shows that it can help escape sharp minima, which is both interesting and meaningful.
- The experiments are fairly comprehensive, covering four mainstream benchmarks and comparing multiple methods and optimizers.

**Weaknesses:**

- As far as I know, the CNC assumption describes the property that determines an optimization algorithm’s ability to escape saddle points. However, a saddle point is not equivalent to a sharp minimum. In fact, [1] shows that the loss landscape of tail classes has a highly negative minimum eigenvalue, indicating convergence to saddle points, which hinders generalization. Therefore, it would be more informative to further examine whether the minimum eigenvalue under Muon is larger than that under SGD, to demonstrate that Muon facilitates escaping saddle points. In Fig. 2, this trend appears plausible, but it would be better to report the exact eigenvalue values for verification.
- More representative long-tailed learning methods, such as GPaCo [2], CC-SAM [3] and DirMixE [4], should be included in the related work for a more comprehensive review.
- There are also some typos, such as in Figure 1, where "Hessain metric" should be "Hessian metric".

-----

[1] Escaping saddle points for effective generalization on class-imbalanced data, NeurIPS 2022

[2] Generalized Parametric Contrastive Learning, TPAMI 2023

[3] Class-Conditional Sharpness-Aware Minimization for Deep Long-Tailed Recognition, CVPR 2023

[4] Harnessing Hierarchical Label Distribution Variations in Test Agnostic Long-tail Recognition, ICML 2024

**Questions:**

Please see above.

---

> ### Author Response · Authors · 2025-11-24
> **Response to Reviewer 7gu8**
>
> > As far as I know, the CNC assumption describes the property that determines an optimization algorithm’s ability to escape saddle points. However, a saddle point is not equivalent to a sharp minimum. In fact, [1] shows that the loss landscape of tail classes has a highly negative minimum eigenvalue, indicating convergence to saddle points, which hinders generalization. Therefore, it would be more informative to further examine whether the minimum eigenvalue under Muon is larger than that under SGD, to demonstrate that Muon facilitates escaping saddle points. In Fig. 2, this trend appears plausible, but it would be better to report the exact eigenvalue values for verification.
>
> Thank you for this insightful comment. We agree that examining the smallest eigenvalue provides a valuable complementary perspective. Following your suggestion, we computed the minimum eigenvalues for the class with the fewest samples under Muon and SGD on both CIFAR-10-LT and CIFAR-100-LT, under an imbalance factor of 100. The results are as follows:
>
> | Dataset      | SGD     | Muon    |
> | ------------ | ------- | ------- |
> | CIFAR-10-LT  | -316.30 | -110.33 |
> | CIFAR-100-LT | -916.12 | -352.22 |
>
> These results show that Muon consistently yields larger minimum eigenvalues than SGD, indicating weaker negative curvature and supporting the view that Muon more effectively escapes saddle-like regions for tail classes. We believe this analysis further clarifies the improved generalization under Muon, and we have included these results in the Experiment section of the revised version.
>
> > More representative long-tailed learning methods, such as GPaCo [2], CC-SAM [3] and DirMixE [4], should be included in the related work for a more comprehensive review.
>
> Thank you for highlighting these representative and advanced works. We agree that including these methods could significantly strengthen the comprehensiveness of our literature review. In the revised version, we have included the following discussion:
>
> Recent advancements in long-tailed recognition have diversified beyond traditional re-balancing techniques. In the realm of contrastive learning, GPaCo [a] identifies the bias of supervised contrastive loss towards high-frequency classes and introduces parametric learnable centers to rebalance optimization dynamics. For handling diverse test distributions, DirMixE [b] proposes a sophisticated mixture-of-experts strategy based on Dirichlet meta-distributions to capture both global and local label distribution variations. Furthermore, addressing the geometry of the loss landscape has become a pivotal direction; CC-SAM [c] argues that naive flattening is insufficient for long-tailed learning and proposes a class-conditional sharpness-aware minimization to robustify the classifier against parameter perturbations.
>
> [a] Generalized Parametric Contrastive Learning, TPAMI 2023.
>
> [b] Harnessing Hierarchical Label Distribution Variations in Test Agnostic Long-tail Recognition, ICML 2024.
>
> [c] Class-Conditional Sharpness-Aware Minimization for Deep Long-Tailed Recognition, CVPR 2023.
>
> > There are also some typos, such as in Figure 1, where "Hessain metric" should be "Hessian metric".
>
> Thanks for pointing this out. We have thoroughly proofread the manuscript and corrected all typos. These changes are included in the revised version.

---

### Official Review · Reviewer_oSTn · 2025-10-31

**Soundness:** 3
**Presentation:** 3
**Contribution:** 2
**Rating:** 4
**Confidence:** 3

**Summary:**

The paper provides a rigorous theoretical analysis of the Muon optimizer, demonstrating its ability to escape sharp minima by enhancing gradient projection along negative curvature directions. This bridges optimization theory with practical long-tailed learning challenges. The proposed ProMO balances computational efficiency and performance, which has been empirically validated on real datasets. In summary, this paper addresses a significant long-tailed learning problem, offering a promising tool for the whole long-tailed learning community. However, the impact of the proposed optimizer on learned representations is somewhat unclear, as the paper focuses more on the efficiency and accuracy issues.

**Strengths:**

1. The proposed ProMO addresses a critical gap in long-tailed learning by balancing computational efficiency and performance. The sinusoidal probability schedule is simple yet empirically effective, outperforming alternatives like linear or exponential schedules.

2. The paper follows a clear structure and is easy to follow. The appendices provide pseudo-code and additional proofs, which enhance reproducibility. The figures and tables are well-designed and support key claims.

3. Evaluation is performed on multiple datasets and loss functions, demonstrating the robustness of the proposed method.

**Weaknesses:**

1. This paper focuses on optimizer design. However, for ICLR, which focuses more on the principles of learning representations, a stronger emphasis on how the method fundamentally improves representation learning beyond just faster convergence is preferable. Although the conference's subject areas include 'large-scale learning and non-convex optimization', my understanding is that it focuses more on how to address the fundamental challenges these issues pose to representation learning, rather than the efficiency limitations imposed by them.

2. The paper does not explore optimizations like gradient approximation or parallelization to further mitigate overhead.

3. The CNC assumption is not validated in a broader context. The claim that Muon’s orthogonalization always enhances negative curvature projection lacks edge-case analysis, e.g., near-convex regions.

4. The source code is not yet publicly available.

**Questions:**

1. The paper focuses on optimizer design. How does Muon fundamentally change feature learning, beyond just better convergence/efficiency compared to the existing optimizers for long-tailed learning?

2. Have the authors compared adaptive methods, e.g., Adam variants?

3. The experiments focus on vision tasks. Is this proposed optimizer promising on NLP or multimodal long-tailed datasets?

---

> ### Author Response · Authors · 2025-11-24
> **Response to Reviewer oSTn (1/3)**
>
> > This paper focuses on optimizer design. However, for ICLR, which focuses more on the principles of learning representations, a stronger emphasis on how the method fundamentally improves representation learning beyond just faster convergence is preferable. Although the conference's subject areas include 'large-scale learning and non-convex optimization', my understanding is that it focuses more on how to address the fundamental challenges these issues pose to representation learning, rather than the efficiency limitations imposed by them.
> > The paper focuses on optimizer design. How does Muon fundamentally change feature learning, beyond just better convergence/efficiency compared to the existing optimizers for long-tailed learning?
>
> Thank you for this thoughtful comment. Our work is motivated by imbalance: in long-tailed recognition, minority classes tend to converge to sharper regions of the loss landscape, which undermines generalization. We make this link explicit in our paper by analyzing loss-landscape geometry in the long-tailed regime and highlighting that the minority-class landscape is often dominated by sharp regions. We respectfully clarify that while many impactful papers regarding optimization (e.g., Adam [a], SAM [b] and their variants [c]) and imbalanced learning (e.g., [d]) were not framed as direct representation analyses, the ICLR community has **historically welcomed** contributions that demonstrably alter learning dynamics and improve generalization.
>
> Methodologically, Muon addresses this **representational failure mode**: its gradient orthogonalization amplifies the update along directions of negative curvature, enabling escape from sharp regions and guiding optimization toward flatter minima. We formalize this with Theorem 1 and support it empirically by reporting loss-landscape metrics on tail classes that indicate flatter solutions relative to SGD, consistent with better generalization in imbalanced settings. In the Discussion section of the revised version, we have summarized this representational perspective and its associated evidence more compactly to focus on how our optimizer addresses the core challenges of representation learning under imbalance.
>
> [a] Adam: A Method for Stochastic Optimization, ICLR 2015.
>
> [b] Sharpness-Aware Minimization for Efficiently Improving Generalization, ICLR 2021.
>
> [c] On the Convergence of Adam and Beyond, ICLR 2018.
>
> [d] Learning to Reject Meets Long-tail Learning, ICLR 2024.
>
> > The paper does not explore optimizations like gradient approximation or parallelization to further mitigate overhead.
>
> Thank you for the constructive suggestion regarding overhead mitigation. To address your concern, we explored gradient approximation by reducing the precision of the Newton–Schulz orthogonalization. Specifically, we further evaluated an approximate version of ProMO on CIFAR-100 LT (IF=100) under CB loss by lowering the iteration steps from N=5 to N=2. The results are shown as follows:
>
> | Method | Steps | Many  | Medium | Few   | All   | Time/epoch  |
> |--------|-------|-------|--------|-------|-------|-------------|
> | SGD    | /     | 75.0  | 50.6   | 17.3  | 44.6  | 2.57(1×)    |
> | Muon   | N= 5  | 76.4  | 53.1   | 19.7  | 46.7  | 4.11(1.59×) |
> | ProMO  | N= 5  | 76.5  | 52.5   | 19.3  | 46.4  | 2.94(1.14×) |
> | ProMO  | N= 2  | 76.1  | 51.1   | 17.7  | 45.2  | 2.59(1.01×) |
> | ProMO  | N= 3  | 76.1  | 52.8   | 18.1  | 45.9  | 2.78(1.09×) |
> | ProMO  | N= 4  | 77.2  | 51.3   | 18.8  | 46.1  | 2.82(1.10×) |
>
> Consistent with our FLOP analysis in Eqs. (6)–(7), reducing the iterations successfully lowers the computational overhead. While the default N=5 retains the highest accuracy, we observe that although the total accuracy decreases slightly as N declines, it consistently remains superior to SGD. This validates that gradient approximation via moderately reduced iterations is still an effective method for maintaining robust performance in resource-constrained scenarios. We included these additional experiments in the Experimental Supplement section of the revised version to provide a more comprehensive analysis of efficiency trade-offs.

---

> ### Author Response · Authors · 2025-11-24
> **Response to Reviewer oSTn (2/3)**
>
> > The CNC assumption is not validated in a broader context. The claim that Muon’s orthogonalization always enhances negative curvature projection lacks edge-case analysis, e.g., near-convex regions.
>
> Thank you for this insightful suggestion. We would like to note that CNC has empirical support across various non-convex settings. For instance, Daneshmand et al. [a] rigorously prove it for learning half-spaces by establishing a **dimension-free** lower bound on gradient variance along the negative curvature. Subsequent works (e.g., [b]) further support CNC through both theoretical analysis and extensive Hessian-spectrum studies on deep networks of varying widths and depths. CNC has been adopted and empirically examined across a broad range of non-convex optimization settings, such as manifold optimization [c], which is why it has become a standard and widely accepted assumption in the literature. Our analysis of Muon builds directly on this established foundation and does not impose any stronger conditions than those required in these prior works.
>
> Crucially for our setting, CNC has also been examined specifically in **long-tailed learning** scenarios. Rangwani et al. [d] conduct class-wise Hessian spectral analysis and show that tail classes exhibit pronounced negative curvature, causing SGD to become trapped in sharp minima. They further demonstrate that methods such as SAM, significantly improve tail-class generalization, offering direct empirical support that CNC is both plausible and practically useful in long-tailed scenarios. Follow-up works, such as [e], reinforce these findings with additional empirical evidence.
>
> To further address your concerns, we additionally perform an edge-case study in a near-convex regime. We consider a quadratic loss $f(w)=\frac{1}{2} w^\top H w$ with a single small negative eigenvalue $\lambda_{\min}\in\\{ -0.1,-0.03,-0.01,-0.003\\} $ and all remaining eigenvalues fixed to one. For each $⁡\lambda_{\min}$, we construct $H=Q\Lambda Q^\top$ with a random orthogonal matrix $Q$, sample $w\sim\mathcal{N}(0,I)$ and gradients $g=Hw$, apply Muon’s Newton–Schulz orthogonalization, and compare the normalized squared projection of the update direction onto the negative-curvature eigenvector before ($p_{\text{SGD}}$) and after ($p_{\text{Muon}}$) orthogonalization. The expectations $\mathbb{E}[p_{\text{SGD}}]$ and $\mathbb{E}[p_{\text{Muon}}]$ are estimated via Monte Carlo over multiple random Hessians and samples:
>
> | $\lambda_{\min}$ | $\mathbb{E}[p_{\text{SGD}}]$ | $\mathbb{E}[p_{\text{Muon}}]$ |
> | ------------------ | --------------------------------- | ----------------------------------- |
> | −0.1           | $7.79\times 10^{-4}$ | $1.89\times 10^{-2}$  |
> | −0.03          | $6.89\times 10^{-5}$ | $1.87\times 10^{-2}$  |
> | −0.01          | $7.67\times 10^{-6}$ | $1.84\times 10^{-2}$  |
> | −0.003         | $6.93\times 10^{-7}$ | $1.81\times 10^{-2}$  |
>
> As $|\lambda_{\min}|$ decreases, the negative eigenvalue suppresses the gradient variance along the negative-curvature direction, so $p_{\text{SGD}}$ quickly vanishes, while Muon’s orthogonalization produces an approximately orthogonal, spectrally normalized update that spreads energy more evenly across directions and keeps $p_{\text{Muon}}$ roughly constant. Consequently, $\mathbb{E}[p_{\text{Muon}}]$ is larger than $\mathbb{E}[p_{\text{SGD}}]$ in the near-convex regime. This confirms that Muon indeed enhances the projection onto negative-curvature directions in these edge cases.
>
> We have revised the Discussion section to more clearly position CNC as a widely used and empirically validated assumption, and incorporated the key references and discussions mentioned above. Thank you again for this insightful comment.
>
> [a] Escaping Saddles with Stochastic Gradients, ICML 2018.
>
> [b] Escaping Saddle Points Faster with Stochastic Momentum, ICLR 2020.
>
> [c] Efficiently Escaping Saddle Points on Manifolds, NeurIPS 2019.
>
> [d] Escaping Saddle Points for Effective Generalization on Class-Imbalanced Data, NeurIPS 2022.
>
> [e] ImbSAM: A Closer Look at Sharpness-Aware Minimization in Class-Imbalanced Recognition, ICCV 2023.
>
> > The source code is not yet publicly available.
>
> Thank you for your valuable feedback. We are fully committed to ensuring that our proposed method is verifiable. To this end, we have included the source code in the supplementary materials. After the anonymous review process, we will release a public repository with more detailed comments, examples, and documentation to further facilitate easy replication of our results.

---

> ### Author Response · Authors · 2025-11-24
> **Response to Reviewer oSTn (3/3)**
>
> > Have the authors compared adaptive methods, e.g., Adam variants?
>
> Thank you for this helpful suggestion. In response, we have carried out additional experiments where we replace SGD with momentum by AdamW while keeping the rest of the training recipe unchanged. Across our long-tailed benchmarks, AdamW consistently yields lower top-1 accuracy than SGD with momentum (typically by about 1–3 percentage points), as in Tab.(a) and (b), so we follow mainstream practice and retain SGD as our default optimizer. This observation is consistent with prior theoretical and empirical work showing that adaptive methods such as Adam can converge faster in optimization but often generalize worse than SGD in image recognition tasks [a].
>
> | Loss | Method | Many | Medium | Few  | All  |
> | ---- | ------ | :--: | :----: | :--: | :--: |
> | CE   | SGD    | 75.6 | 62.8   | 48.2 | 60.8 |
> | CE   | AdamW  | 74.2 | 62.0   | 47.3 | 59.8 |
> | LA   | SGD    | 70.0 | 64.3   | 57.1 | 63.2 |
> | LA   | AdamW  | 70.5 | 63.4   | 53.3 | 61.5 |
>
> *Tab.(a) Comparison on CIFAR-100 LT under IF=10.*
>
> | Loss | Method | Many | Medium | Few  | All  |
> | ---- | ------ | :--: | :----: | :--: | :--: |
> | CE   | SGD    | 75.9 | 52.0   | 15.7 | 44.6 |
> | CE   | AdamW  | 73.3 | 46.6   | 14.7 | 41.9 |
> | LA   | SGD    | 69.2 | 53.6   | 34.3 | 50.5 |
> | LA   | AdamW  | 68.2 | 50.8   | 29.2 | 47.4 |
>
> *Tab.(b) Comparison on CIFAR-100 LT under IF=100.*
>
> Moreover, our choice is aligned with the established practice in long-tailed recognition. Most advanced methods on mainstream long-tailed benchmarks, including BCL and ProCo, train their models with SGD-style optimizers and learning-rate schedules such as cosine decay, rather than Adam-type methods, in order to achieve better performance. Furthermore, instead of focusing on generic adaptive methods, our work targets the optimization challenge of long-tailed learning more directly by comparing optimizers that are specifically effective for improving generalization in this setting, such as SAM. As detailed in the paper, we evaluate four optimizers (SGD, SAM, Muon, and ProMO) across multiple long-tailed losses (such as BCL and ProCo) and datasets, and show that Muon and ProMO consistently outperform both SGD and SAM, especially on tail classes, while maintaining reasonable computational cost. We believe this analysis is more informative for the long-tailed setting, and thanks again for your insightful suggestion that has enriched our empirical analysis.
>
> [a] The Marginal Value of Adaptive Gradient Methods in Machine Learning, NeurIPS 2017.
>
> > The experiments focus on vision tasks. Is this proposed optimizer promising on NLP or multimodal long-tailed datasets?
>
> Thank you for the suggestion to evaluate our method beyond vision tasks. While we initially followed mainstream long-tailed learning protocols, such as [a],  which mainly focus on visual benchmarks, we agree that demonstrating potential in other modalities is valuable. To address this concern, we conducted supplementary experiments in NLP using the Yahoo Answers Topic Classification dataset. Specifically, we constructed long-tailed versions of the dataset by sampling from a 12k training subset with imbalance factors (IF) of 10 and 50, alongside a balanced 4k test set. We employed a fixed pre-trained BERT-base-uncased backbone followed by an MLP layer and a linear classification head, training with CE loss. Comparing the accuracy of SGD and Muon over 20 epochs, the results (shown in the table below) demonstrate that Muon consistently outperforms SGD, achieving superior generalization performance across different imbalance factors. This shows that Muon's effectiveness extends to broader domains. We have included the experiment details in the Experimental Supplement section of the revised version.
>
> | IF | Method | Many  | Medium  | Few   | All   |
> |----|--------|:-----:|:-----:|:-----:|:-----:|
> | 10 | SGD    | 75.9  | 58.5  | 43.3  | 59.1  |
> | 10 | Muon   | 73.0  | 60.9  | 45.6  | 59.9  |
> | 50 | SGD    | 74.8  | 61.2  | 3.4   | 47.9  |
> | 50 | Muon   | 76.8  | 55.9  | 17.3  | 50.6  |
>
> [a] Long-tail Learning via Logit Adjustment, ICLR 2021.

---

> > ### Comment · Reviewer_oSTn · 2025-11-25
> > **The authors' rebuttal has addressed my concerns, I will increase my rating.**
> >
> > I appreciate the authos' careful and comprehensive responses.
> > After reading the other reviewers' comments and the rebuttal, I think that my concerns have been well addressed.
> > I will raise the score.

---

### Comment · Area_Chair_F1SH · 2025-11-22

Dear Reviewers,

Thank you for your time and effort in reviewing submissions for ICLR  2026. As we begin the author-reviewer discussion process, we kindly remind you to submit your responses to the author rebuttals by **December  2**.


Your engagement in this discussion phase is crucial to ensuring a fair and thorough evaluation of each submission.

**Action Required**


- Carefully consider the authors’ rebuttal and any additional evidence they provide.

- Update your review (if applicable) to reflect your revised perspective.

-  **Discuss with the authors if further details are required**


Your AC

---

### Public Comment · ~Jialong_Liu3 · 2025-11-26
**Question About Theorem 1**

Hi Authors,

Thanks for bringing this interesting paper to the community. However, I was **unable to fully reconstruct the proof of Theorem 1** from the arguments given in the paper. In particular, if I understand correctly, the proof of Theorem 1 relies crucially on **an implicit assumption in Line 987: “We focus on the late stages of training near convergence, where the gradient norm becomes very small.”** Under this assumption, my understanding is that $\gamma$ should be arbitrarily small. In that regime, Theorem 1 effectively states that when the gradient itself is arbitrarily small, Muon normalizes all singular values of the update to $1$, so the Muon update has a larger norm. **This conclusion, however, appears to be vacuous for the practically relevant regime where the gradient norm is not necessarily small (e.g., at earlier stages of training or when the gradient has medium or large norm).**

I would greatly appreciate it if the authors could help resolve my question.

Best,
A reader

---

> ### Author Response · Authors · 2025-12-03
>
> We thank you for the detailed comment. We would like to clarify that the regime where the gradient norm is small is not merely a trivial case, but rather the critical bottleneck in long-tailed learning that our paper aims to address.
>
> In long-tailed scenarios, tail classes tend to become trapped in sharp local minima or saddle points during the late stages of training. In these sharp regions, the gradient update of SGD vanishes, causing the optimization to stall, as discussed in works such as [1]. Theorem 1 formalizes that precisely in this regime, where SGD fails to provide sufficient movement, Muon normalizes the singular values to maintain a significant update magnitude. Therefore, the result is not vacuous; it mathematically characterizes Muon’s unique capability to improve optimization dynamics and escape sharp regions exactly when SGD loses its efficacy.
>
> This insight serves as the technical foundation for ProMO. We posit that while SGD is sufficient during early exploration when gradients are large, Muon becomes essential during late-stage convergence to escape sharp minima. To empirically validate this dynamics-based motivation, we tracked the Hessian trace of the least frequent class on CIFAR-100 LT under IF=100 throughout training. The results are presented below:
>
> | Epoch | SGD               | Muon               | ProMO               |
> | :---- | :---------------- | :----------------- | :------------------ |
> | 80    | 664.6             | 636.1              | 691.5               |
> | 120   | 956.1             | 450.8              | 615.2               |
> | 180   | 1629.6            | 703.6              | 667.6               |
> | 200   | 2137.5            | 536.0              | 514.1               |
>
> As shown in the table, both SGD and Muon maintain relatively low sharpness in the early stages. However, as the gradient norm diminishes in the later stages, the Hessian trace for SGD spikes dramatically, confirming its entrapment in sharp minima. In contrast, Muon and ProMO maintain a flat geometry. This confirms that the theoretical regime of small gradients in Theorem 1 is critical for optimization: it corresponds to the decisive phase where Muon effectively prevents the sharpness spikes, thereby capturing the essential mechanism for improving generalization on tail classes.
>
> [1] Sharpness-Aware Minimization Efficiently Selects Flatter Minima Late in Training, ICLR 2025.

---

### Author Response · Authors · 2025-12-03
**Summary and Clarification of the Rebuttal**

Dear Area Chairs,

We would like to express our deep gratitude for your valuable time and effort in handling our submission, especially given the information leakage incident, which brings more burden to the Area Chairs. To ease the labor in the review checking process and clarify the critical details in the rebuttal, we prepared a summary below as a reference for your evaluation.


### Summary on Strengths
We appreciate that the reviewers recognized the novelty of connecting gradient orthogonalization to long-tailed learning and the practical value and the effectiveness of our method. Initially, our submission received scores of 6, 6, 4, 4 (avg. 5.0), and during the rebuttal, we achieved **6, 6, 6, 4 (avg. 5.5)** after valid discussion, which happened before the information leakage incident. Specifically, **Reviewer oSTn (score 4 to 6) and ZKRh (score 6) acknowledged that concerns had been well addressed, and the former raised the score from 4 to 6 on Nov 25 UTC**. We engaged in a constructive dialogue with Reviewer MTSZ (score 4) until the discussion concluded unexpectedly early, after which we provided further responses to the remaining questions and believe these clarifications would have been similarly well-received.

In the following, we listed the consensus from the reviewers on the key strengths of our work:

- **Novel theoretical insight**: Reviewers (oSTn and ZKRh) acknowledge our valuable theoretical analysis, highlighting that connecting gradient orthogonalization to escaping sharp minima provides a sound foundation for long-tailed learning.
- **Effective and practical design**: Reviewers (oSTn and MTSZ) recognize ProMO as a simple yet technically sound solution that effectively balances computational efficiency with robust generalization.
- **Comprehensive evaluation**: Reviewers (oSTn, 7gu8, and ZKRh) praise our extensive experiments across diverse benchmarks and loss functions, demonstrating the method's robustness and superior performance.
- **High-quality presentation**: Reviewers (oSTn, 7gu8, and MTSZ) commend the paper's clarity and structure, noting that the logical flow makes the technical contributions easy to follow.

### Rebuttal Actions
During the rebuttal and discussion, we have provided critical clarifications and conducted comprehensive extra experiments to solidify our contributions:

- **Additional comparisons with advanced baselines.** To demonstrate superiority and robustness, we added results on **large-scale** iNaturalist-2018 (suggested by MTSZ) and **NLP tasks** (oSTn), showing consistent improvements over baselines. We further compared against **strong SAM variants** (ImbSAM, LookSAM), proving ProMO achieves higher accuracy with significantly lower cost (e.g., 1.29x vs. ImbSAM’s 3.09x) (oSTn), and validated effectiveness in diverse settings including **Fine-tuning** and **Decoupled Training** scenarios (ZKRh) (added in Appendix C.4-C.8).

- **Deeper theoretical and geometric analysis.** We clarified the underlying training dynamics by reporting minimum eigenvalues to confirm ProMO’s convergence to flatter minima with weaker negative curvature compared to SGD (7gu8). We also conducted an **edge-case study** in near-convex regions to further substantiate the CNC assumption (oSTn). Additionally, to strengthen the connection between our **theoretical analysis and algorithmic design**, we tracked Hessian trace evolution, validating ProMO’s ability to escape sharp regions via sinusoidal scheduling (MTSZ) (added in Appendix C.9).

- **Efficiency, scalability, and reproducibility.** We reinforced practical value by demonstrating that **gradient approximation** via reduced Newton-Schulz steps mitigates overhead while maintaining robust performance in resource-constrained scenarios (oSTn). Furthermore, we enhanced reproducibility by including the source code, thoroughly proofreading the manuscript, and expanding the Related Work section (e.g., GPaCo, DirMixE)  (added in Appendix C.3 and D).

Above improvement makes our contribution stronger and clearer, establishing our method as a robust and efficient solution for long-tailed learning. We deeply value all reviewers' feedback and sincerely appreciate the time and efforts of the Area Chairs. Hope the summary above provide a clear reference for your judgement and evaluation.

Best regards,

The Authors of Submission 10542

---

### Meta-Review · Area_Chair_bCVJ · 2026-01-01

**Summary:**

The paper introduces ProMO, a hybrid optimization framework that progressively integrates the Muon optimizer’s gradient orthogonalization into a standard SGD pipeline. The goal is to help models escape the sharp minima that characterize tail-class loss landscapes in long-tailed learning scenarios.

The Area Chair acknowledges the authors' exemplary responsiveness during the rebuttal. The addition of results on iNaturalist-2018, NLP tasks, and the tracking of Hessian traces provided valuable empirical support for the underlying motivation. However, despite these additions, the current version of the paper does not yet meet the high threshold for an ICLR publication. The primary reasons for rejection are the largely heuristic nature of the proposed scheduling mechanism and the lack of competitive performance on large-scale benchmarks.

**Reviewer Concerns:**

**Addressed by the Rebuttal:**
*   The authors successfully validated the "sharp minima" hypothesis for tail classes through Hessian spectral analysis. This provides a solid motivation for using second-order-like optimizers like Muon.
*   The authors demonstrated that ProMO offers a superior accuracy-efficiency trade-off compared to SAM-based methods (e.g., LookSAM), which is a valuable practical insight.
*   The inclusion of NLP and fine-tuning experiments demonstrated the potential versatility of the Muon-based approach.

**Outstanding Concerns (Reasons for Rejection):**
*   While the authors prove that Muon’s orthogonalization helps escape sharp minima, the bridge to the **sinusoidal probability schedule** remains heuristic. Here, the specific choice of a sinusoidal curve over other schedules (linear, exponential, or loss-adaptive) lacks a rigorous theoretical or empirical derivation. To be convincing, the paper needs to demonstrate that this specific schedule is a principled requirement of the optimization dynamics, rather than a tuned hyperparameter.
*   The iNaturalist-2018 results provided during the rebuttal, while showing improvement over baseline SGD, are significantly below the current SOTA in the long-tailed learning field. For a method to be established as a "robust and efficient solution" for LTL, it must demonstrate that it can either achieve SOTA or be combined with existing SOTA methods to push the frontier further. Currently, the performance gap suggests the method may be an auxiliary improvement rather than a foundational advancement in LTL.
*   Reviewers noted that the paper remains focused on optimization mechanics. A more compelling version of this work would analyze how escaping these sharp minima specifically changes the **learned feature representations** of tail classes.

**Reviewer Scores:**

*   **Reviewer oSTn:** **Initial: 4 $\rightarrow$ Final: 6.**
*   **Reviewer 7gu8:** **Initial: 6 $\rightarrow$ Final: 6.**
*   **Reviewer ZKRh:** **Initial: 6 $\rightarrow$ Final: 6.**
*   **Reviewer MTSZ:** **Initial: 4 $\rightarrow$ Final: 4.**

---

### Decision · Program_Chairs · 2026-01-26

Reject